# Iron rescues glucose-mediated photosynthesis repression during lipid accumulation in the green alga *Chromochloris zofingiensis*

Tim L. Jeffers[1], Samuel O. Purvine [2], Carrie D. Nicora [3], Ryan McCombs [1], Shivani Upadhyaya [1], Adrien Stroumza[1], Ken Whang[1], Sean D. Gallaher [4,5], Alice Dohnalkova [2], Sabeeha S. Merchant [1,5,6,7], Mary Lipton [2], Krishna K. Niyogi [1,8,9] ✉ & Melissa S. Roth [1] ✉

Energy status and nutrients regulate photosynthetic protein expression. The unicellular green alga *Chromochloris zofingiensis* switches off photosynthesis in the presence of exogenous glucose (+Glc) in a process that depends on hexokinase (HXK1). Here, we show that this response requires that cells lack sufficient iron (−Fe). Cells grown in −Fe+Glc accumulate triacylglycerol (TAG) while losing photosynthesis and thylakoid membranes. However, cells with an iron supplement (+Fe+Glc) maintain photosynthesis and thylakoids while still accumulating TAG. Proteomic analysis shows that known photosynthetic proteins are most depleted in heterotrophy, alongside hundreds of uncharacterized, conserved proteins. Photosynthesis repression is associated with enzyme and transporter regulation that redirects iron resources to (a) respiratory instead of photosynthetic complexes and (b) a ferredoxin-dependent desaturase pathway supporting TAG accumulation rather than thylakoid lipid synthesis. Combining insights from diverse organisms from green algae to vascular plants, we show how iron and trophic constraints on metabolism aid gene discovery for photosynthesis and biofuel production.

Photosynthesis is a major driver of the global carbon cycle and nearly all ecological food webs. While the electron transport chain (ETC) in oxygenic photosynthesis is conserved across distantly related organisms[1], individual species have evolved diverse trophic strategies and mechanisms of photosynthetic regulation and carbon storage. For example, several green algae supplement photosynthetic metabolism through mixotrophy (i.e., autotrophy + heterotrophy), which includes consuming external carbon metabolites[2–4].

Mixotrophy has been exploited by geneticists to unveil essential players in photosynthesis. For example, non-photosynthetic mutants of the reference green alga *Chlamydomonas reinhardtii*, which survive on acetate as a sole carbon source, reveal genes required for

[1]Department of Plant and Microbial Biology, University of California, Berkeley, CA 94720, USA. [2]Environmental Molecular Sciences Laboratory, Pacific Northwest National Laboratory, Richland, WA 99354, USA. [3]Biological Sciences Division, Pacific Northwest National Laboratory, Richland, WA 99354, USA. [4]UCLA DOE Institute for Genomics and Proteomics, University of California, Los Angeles, CA 90095, USA. [5]Quantitative Biosciences Institute, University of California, Berkeley, CA 94720, USA. [6]Department of Molecular and Cell Biology, University of California, Berkeley, CA 94720, USA. [7]Environmental Genomics and Systems Biology Division, Lawrence Berkeley National Laboratory, Berkeley, CA 94720, USA. [8]Howard Hughes Medical Institute, University of California, Berkeley, CA 94720-3102, USA. [9]Molecular Biophysics and Integrated Bioimaging Division, Lawrence Berkeley National Laboratory, Berkeley, CA 94720, USA. ✉e-mail: niyogi@berkeley.edu; mroth@berkeley.edu

photosynthesis[5–7]. Lipid accumulation in algae is additionally studied to develop sustainable biofuels. Green algae, including *C. reinhardtii*, store large quantities of triacylglycerols (TAGs), ideal precursors for biodiesel, in growth-inhibiting conditions[8]. Multi-omics analyses of nutrient deprivation have elucidated metabolic routes for increasing TAG production[9]. However, multi-omics analysis of algae such as *Chromochloris zofingiensis*, which accumulate TAG while increasing biomass, could provide diverse strategies to engineer biofuels.

An emerging reference green alga with high-quality genome annotations, *C. zofingiensis* regulates photosynthesis and TAG accumulation by unique mechanisms[10–12]. *C. zofingiensis*, unlike *C. reinhardtii*, naturally consumes exogenous glucose (+Glc), which can trigger a complete switch-off of photosynthesis in the light and TAG accumulation with increasing biomass[12,13]. During the photosynthetic switch-off, thylakoids and photosystems are depleted[12]. In TAG accumulation mediated by +Glc or nutrient deprivation, *C. zofingiensis* rapidly upregulates transcripts associated with de novo fatty acid synthesis (*dn*FAS), thereby increasing total fatty acids available for TAG production[10,12]. This metabolic and transcriptional mechanism differs from *C. reinhardtii*, which downregulates *dn*FAS during TAG accumulation and relies on membrane remodeling to support TAG accumulation[9,14].

Glc-mediated repression of photosynthesis and promotion of TAG accumulation require *C. zofingiensis'* single-copy hexokinase gene (*CzHXK1*), which encodes the first enzyme in glycolysis[13]. As in plants, fungi, and animals, HXK1 in *C. zofingiensis* mediates a transcriptomic response to Glc[13,15,16]. In plants, many Glc responses are independent of HXK signaling[15,17,18], and further challenges remain across organisms in distinguishing whether +Glc signaling responses are mediated by HXK Glc sensing vs. Glc phosphorylation[17].

In this study, we reveal insufficient iron (−Fe) is a third factor that is essential for switching off photosynthesis in *C. zofingiensis*. Iron is an essential cofactor in both photosynthetic and respiratory ETCs[19]. Iron limitation across organisms impedes optimum physiology[20,21] causing cells to spare non-essential iron-containing proteins to prioritize "essential" iron cofactor enzymes[19,22]. During iron limitation in plants and algae, the iron-containing enzymes of photosynthesis are spared in favor of maintaining the respiratory ETC[23–27]. In *C. reinhardtii*, the negative effects of iron limitation on photosynthesis increase synergistically in combination with acetate, likely because acetate stimulates respiration, although photosynthesis is not completely shut off[24,27–29]. Because mRNA levels often do not predict proteomic responses of ETC subunits[12,21,30,31] or iron-cofactor proteins[32], quantitative proteomics is an ideal technique to investigate ETC and cofactor regulation[25,33,34].

Here, we show that wild-type *C. zofingiensis* maintains photosynthesis with Glc when high iron is supplemented (WT+Fe+Glc). Microscopy and analytical chemistry confirmed that WT+Fe+Glc cells maintain thylakoids, while +Glc via HXK1 causes TAG accumulation regardless of Fe state. Because three factors are essential for switching off photosynthesis: +Glc, HXK1, and insufficient Fe (−Fe), we used proteomics to distinguish photosynthesis vs. lipid accumulation responses. We designed a full factorial proteomics experiment: 12 conditions for all combinations of Fe (+/−) and Glc (+/−) in WT and two *hxk1* mutant strains. Our statistical categorization by linear models showed more proteins significantly respond to additive or synergistic effects of Fe and Glc ("*Fe:Glc*") than to each of these inputs individually. This design allowed us to distinguish protein signatures of photosynthesis repression from those involved in growth inhibition and TAG accumulation, which often occur under the same conditions in other experiments[10,12,35]. Comparing these candidate proteins to their orthologs in other algae and plants identified hundreds of experimentally uncharacterized photosynthesis-associated proteins that are conserved across Viridiplantae. Finally, proteomic evidence shows iron was prioritized to respiration over photosynthesis. Unlike other algae, *C. zofingiensis* showed iron prioritization to a subset of iron-containing

enzymes, especially ferredoxins (FDXs) and fatty acid desaturases (FADs), that support TAG accumulation at the expense of thylakoid biogenesis.

## Results

### Iron deficiency is essential for glucose-mediated photosynthesis repression via HXK1

*C. zofingiensis* grown in Proteose medium (iron concentration is undefined) switched off photosynthesis after treatment with exogenous Glc (35 mM) in the light, consistent with previous experiments[12,13] (loss of detectable maximum quantum efficiency of photosystem (PS) II $F_v/F_m$ Fig. 1a, b). However, when an iron (10 μM) supplement (+Fe) was added concurrently with +Glc, photosynthesis was maintained (Fig. 1a, b, Supplementary Data 1). In −Glc cultures, +Fe induced a small but statistically significant increase in $F_v/F_m$ (Fig. 1a), indicating that cells were iron deficient when grown under these conditions. +Glc increased the volumetric biomass >2.7-fold at 72 h in both Fe regimes compared to −Glc (Fig. 1d, Supplementary Fig. 2a). Moreover, +Fe+Glc further increased biomass ~4-fold, but +Fe did not improve −Glc growth, showing glucose affects growth more than Fe supplementation (Fig. 1c, d, Supplementary Data 1). Within 84 h, in −Fe+Glc and +Fe +Glc ~60% and 80% of glucose was consumed, respectively. −Fe−Glc cultures continued to grow at similar rates as +Fe−Glc despite treatment (Fig. 1d, Supplementary Fig. 2b) indicating low iron did not reduce or cease growth. Combined photosynthesis with Glc consumption (mixotrophy) may contribute to the +Fe+Glc biomass increases, although it is unlikely to explain the 4-fold increase. Altogether, these data show that in +Glc, −Fe inhibited photosynthesis and the maximum biomass gain.

Because the initial growth medium was iron deficient, we confirmed that no other nutrient supplement maintained photosynthesis in +Glc. A survey of mineral nutrient supplements (nitrogen, phosphorus, magnesium and sulfur, manganese, copper, and zinc) with +Glc, showed that only +Fe+Glc treatments maintained photosynthesis ($F_v/F_m$) (Supplementary Fig. 1a). The increase in +Glc volumetric biomass also occurred only in +Fe+Glc (Supplementary Fig. 1b). These results indicate that −Fe was the primary nutritional limitation to +Glc growth, and −Fe was likely the only nutrient limitation feature of this medium that permitted Glc-mediated repression of photosynthesis.

In addition to metabolic differences, −Fe+Glc and +Fe+Glc cells were morphologically distinct. Light microscopy of representative cells showed that +Glc increased cell size in both Fe states, but +Fe+Glc cells were ~2.7 times larger than −Fe+Glc cells by mean cellular volume (Fig. 2a). Transmission electron microscopy (TEM) of cells showed the impact of trophic state on plastid morphology. In whole-cell images, starch appeared more prevalent inside +Glc plastids, but +Fe+Glc had the largest starch granules (Fig. 2b, Supplementary Data 2). Quantitative starch measurements confirmed +Glc cultures contained more starch than −Glc cultures, and that +Fe+Glc cultures had twofold more starch than −Fe+Glc (Supplementary Fig. 2c). More starch in +Fe+Glc relative to −Fe+Glc may result from mixotrophic carbon incorporation via both photosynthesis and Glc metabolism. Consistent with the loss of photosynthesis, it was very difficult to detect thylakoid membranes in heterotrophic cells (−Fe+Glc), a treatment where ~14-fold reduction in thylakoid membranes was previously quantified[12]. In contrast, thylakoids were easily detectable in mixotrophic cells (+Fe+Glc) and in both photoautotrophic conditions (+Fe−Glc or −Fe−Glc, Fig. 2c). Therefore, +Fe+Glc cells maintained both photosynthesis and thylakoid membranes.

TAG accumulation during growth in +Glc occurred independently of photosynthesis loss in WT cells. Thin-layer chromatography (TLC) (Fig. 2d) showed TAG accumulation in both heterotrophic (−Fe+Glc) and mixotrophic (+Fe+Glc) cells of WT. Therefore, +Glc enhanced TAG accumulation regardless of iron status, whereas switching off photosynthesis occurred specifically by the combination of −Fe+Glc in WT.

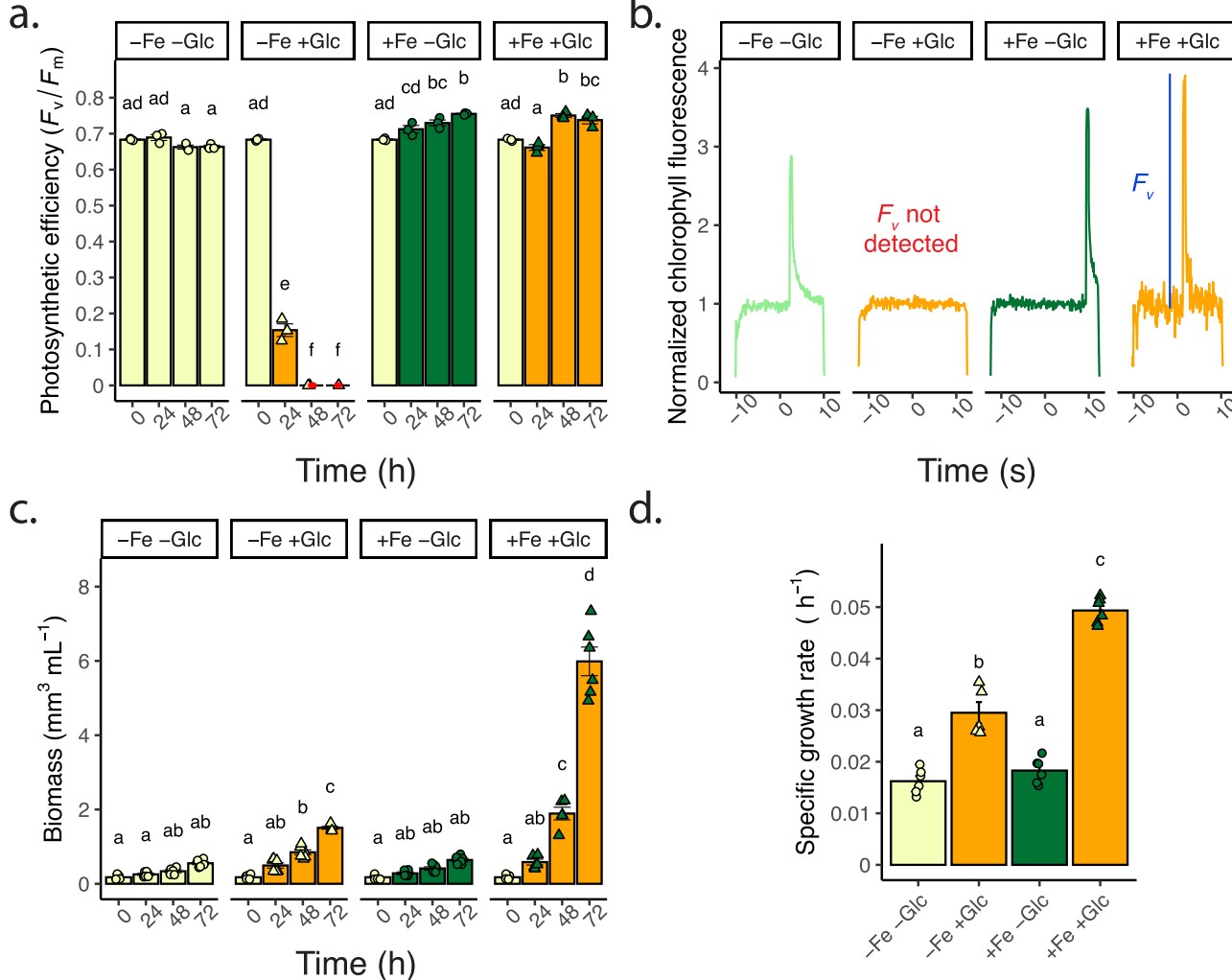

**Fig. 1 | Replete iron rescues glucose-mediated repression of photosynthesis in WT. a** Maximum quantum efficiency of PSII ($F_v/F_m$) with Glc and Fe treatments over 72 h. $F_v$ signal not detected outside of $F_o$ noise, $F_v/F_m$ was assigned as 0 (indicated with a red dot). Data represent means ± SE ($n = 3$, individual data points shown, from one experimental replicate). **b** Representative individual sample chlorophyll fluorescence (normalized to median signal) of saturating light. −Fe+Glc sample (samples 48 h, orange line) had no detectable $F_v$ during saturating light induction compared to $F_o$ noise (measuring light fluorescence). *x*-axis is time normalized by mean time point of sample load and run. **c** Volumetric algal biomass (per mL of culture) over 72 h. Data represent means ± SE ($n = 6$, individual data points shown from two experimental replicates). **d** Specific growth rate of volumetric biomass of time courses in (**c**). Data represent means ± SE ($n = 5–6$, individual data points shown from two experimental replicates). Statistical comparisons, represented by letters, are applied by Tukey's HSD with adjusted $p = 0.05$. Source data are provided in the Source Data File.

## Iron and glucose interact to shape the global proteomic response

To investigate the interactions of Glc, HXK1, and Fe in controlling metabolism in *C. zofingiensis*, we used proteomic signatures to discriminate the photosynthetic switch-off from lipid accumulation and nutrient signaling. We included *hxk1* mutants, which do not switch off photosynthesis nor accumulate lipids in +Glc[13], to distinguish general +Glc response from those that are HXK1-dependent and induce heterotrophy (−Fe+Glc) or TAG accumulation. +Fe in WT+Glc cultures induces TAG accumulation without photosynthetic loss, further enhancing the ability to separate TAG vs. photosynthesis signatures. Our full-factorial experimental design included 12 conditions of +/−Fe, +/−Glc, and WT vs. two independent *hxk1* strain combinations for quantitative proteomic mass spectrometry by tandem mass tag labeling (Fig. 3a). Mid-log WT, *hxk1-1*, and *hxk1-2* cultures were treated with +/−Fe and +/−Glc and collected for mass spectrometry at 84 h, when WT−Fe+Glc cells were heterotrophic[12,13] (Fig. 3a). Offline protein fractionation before the standard mass-spectrometry protocol (Methods, Supplementary Information) provided high proteomic coverage: 10,100/14,068 (~72%) annotated *C. zofingiensis* gene models[36] had ≥2 unique detected peptides across the experiment. Of these, 7,258/14,068 (~55%) were quantified in all 12 conditions (Fig. 3b). Principal Components (PC) analysis (Fig. 3c) and autocorrelation analysis (Supplementary Fig. 3a) of the 55% of proteins showed high replicability within conditions. Fe and Glc treatments were clearly split by PC1 (accounting for 22.0% variation) and PC2 (17.6%), respectively. Proteomic differences from strain identity were not clearly separated until PC4 (10.8%) and PC8 (4.2%, Supplementary Fig. 3b).

The *hxk1-1* proteome correlated less well with WT or *hxk1-2* than WT and *hxk1-2* correlated with each other (Supplementary Fig. 3a). This divergence could be due to unique polymorphisms in *hxk1-1*[13]. Therefore, *hxk1-1* was removed from the linear model classification pipeline (below) but was still used for targeted photosynthesis and lipid accumulation analysis, where both *hxk1* strains show the same metabolic phenotypes.

To statistically categorize how nutrients and strain identity shape protein levels, we modeled each protein's abundance in a full factorial linear model including WT vs. *hxk1-2* (Fig. 3d, Methods, Supplementary

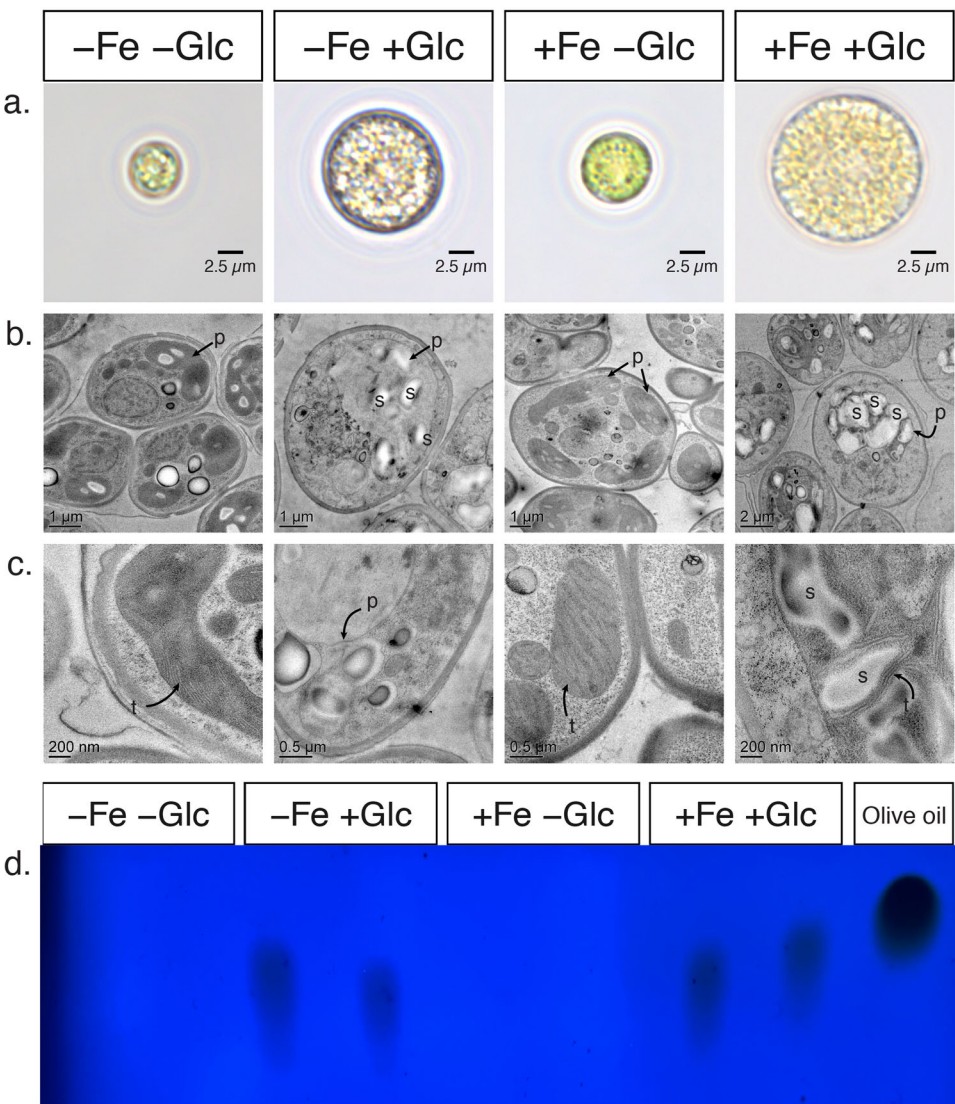

**Fig. 2 | Mixotrophic (WT+Fe+Glc) cells are larger, maintain thylakoid membranes and accumulate starch. a** Light microscopy of cells with Glc and Fe at 84 h. **b** Transmission electron microscopy (TEM) of whole cells at 96 h showing plastid (**p**) and starch granules (**s**). Note that cells were in the dark (during shipping) for ~24 h prior to sampling for TEM so starch differences may have been greater at 84 h. **c** Detailed TEM of plastid and thylakoids (**t**), which appear as stacked membranes in −Fe−Glc, +Fe+Glc and +Fe+Glc. Thylakoids in −Fe+Glc are small, fragmented, and difficult to see in −Fe+Glc cultures (see ref. 12. for a 3D reconstruction of thylakoid membranes). **d** UV-Vis TLC plates of lipid extracted from equal volumetric biomass of Fe and Glc cultures at 84 h and an olive oil TAG standard. Two biological replicates are loaded per treatment and an experimental replicate can be found in Fig. 7a. A complete set of images can be found in Supplementary Data 2 or on the online data repository https://osf.io/r8dbe/. Source data are provided in the Source Data File.

Data 3). To generalize protein responses into simplified models, a computational filtering pipeline reduced each linear equation to its major effect terms (e.g., *Fe, Glc* or *Fe:Glc* effects) that had statistically significant and dominant influence across conditions. We evaluated the accuracy of this statistical strategy based on its classification of well-characterized algal biomarkers of iron deficiency (FLD1, FOX1, FRE1, FEA1, and FEA2)[37–39]. Consistent with their function, all five proteins' simplified equations captured their expected upregulation by −Fe (Supplementary Fig. 4a, b). The modeling pipeline did not cause overfitting, as confirmed through cross-validation (Methods, Supplementary Fig. 4c).

The major protein groups from this categorization analysis are shown as heatmaps in Fig. 3e, where the *y*-axis represents number of proteins with shared statistical terms (*x*-axis). The highest number of proteins were described by three effect terms ("*Fe, Glc, Fe:Glc*") and were up- or downregulated, regardless of strain, only in one of the four Fe and Glc combinations. The largest subgroup described by these

terms, "+*Fe* +*Glc* + *Fe:Glc*", where each term is positive, comprises proteins that are significantly upregulated only in WT+Fe+Glc and *hxk1-2* + Fe+Glc. Like this subgroup, most proteins are characterized by significant responses to both "*Fe*" and "*Glc*", and/or their synergy "*Fe:Glc*", rather than to a single input. This result suggests that proteomic Fe responses are generally influenced by Glc state (or vice versa).

The categorization by significant effect terms and the PC plot showed that Fe and Glc have a much larger effect on the proteome than strain effects alone. The HXK1 protein was found in the small subgroup "+*strain*", where abundance was always higher in WT than in *hxk1-2*. Despite the presence of frameshift mutation causing an early stop codon, which is shared near the N-terminus by both strains, we detected wild-type HXK1-aligned peptides in both *hxk1* strains in all conditions at a ~ 4.5X lower abundance compared to WT (Supplementary Fig. 5a). 43/49 unique detected peptides in the *hxk1* strains map downstream of the early stop codon[13] (Supplementary Fig. 5b–d),

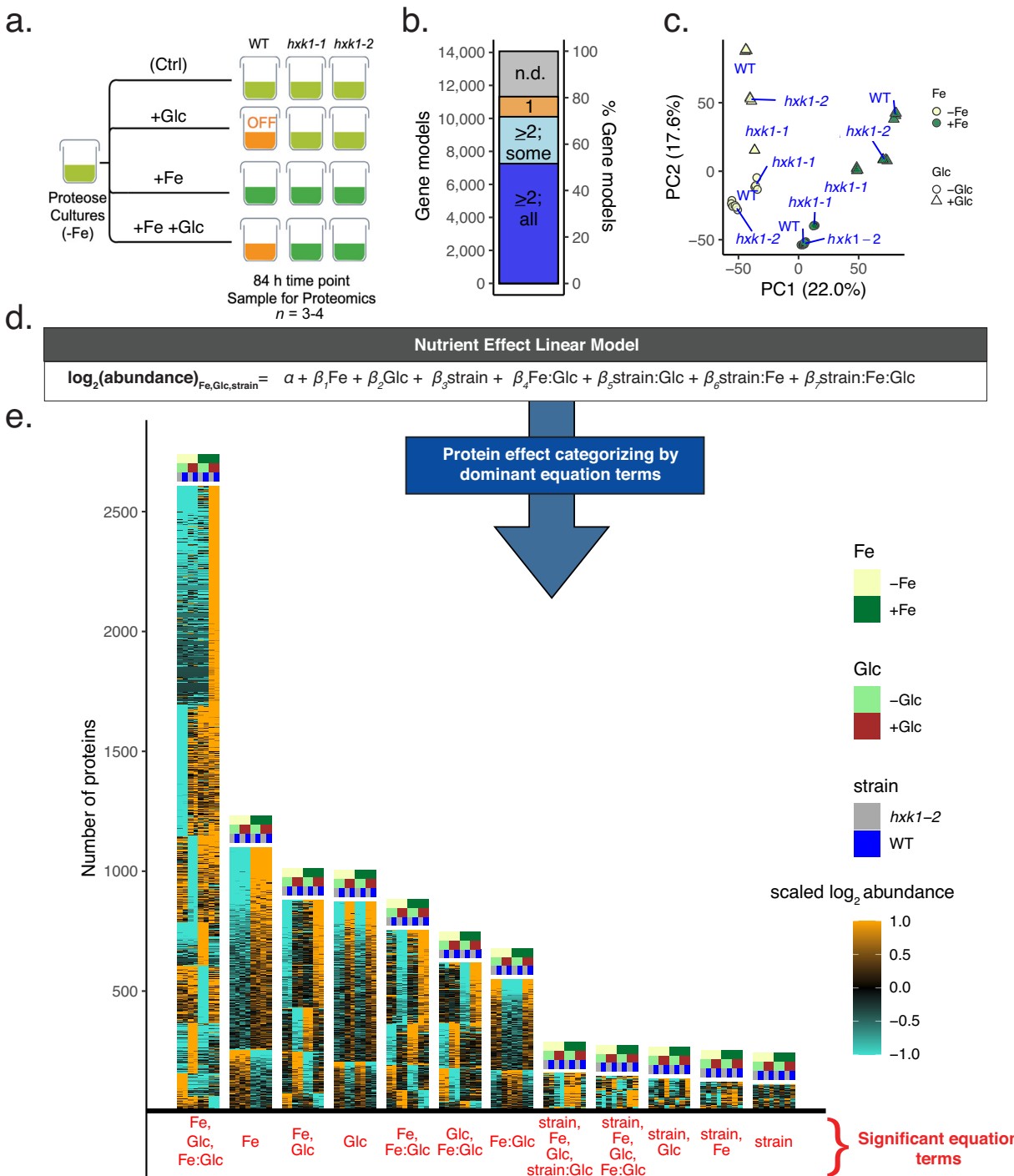

**Fig. 3 | A full-factorial proteomic design reveals prominence of synergistic effects on differential protein abundance. a** Experimental design of WT, *hxk1-1* and *hxk1-2* given all 12 combinations of +/- Fe and +/Glc (*n* = 3–4; see Supplementary Fig. 6a for culture photographs). Proteomics was conducted on treatment replicates from the same experimental batch. **b** Number of gene models (*y*-axis, left) and % of gene models (right) with TMT-labeled peptides. Bar colors indicate genes ≥2 unique representative peptides and detected in all conditions (blue), genes with ≥2 unique representative peptides but are undetected in some conditions (light blue), genes with peptides with just one detected peptide (tan), and genes with no detected peptides (n.d., gray). **c** Principal components analysis of peptides detected in all treatments. **d** Full linear model tested against proteins as a function of Fe, Glc, and strain (WT vs. *hxk1-2*) (see Methods). **e** Results of a data-filtering pipeline to categorize proteins by the variables that have significant impacts on their proteome (Supplementary Data 3). Height of each heatmap corresponds to the number of proteins (*y*-axis) associated with significant linear model terms (*x*-axis). Source data are provided in the Source Data File.

suggesting that the frameshift mutation confers a knock-down phenotype in both *hxk1* strains. This result is consistent with research on the molecular consequences of frameshift mutations, where various mechanisms allow for translation downstream of predicted early stop codons[40,41].

Since most "*+Glc*" responses in *C. zofingiensis* were designated strain-independent, they may be part of HXK-independent responses like those previously described in plant *hxk* mutants[15,16]. Strain-independent responses may be related to cell size, which we found increases in all strains and iron conditions with +Glc (Supplementary

Fig. 2a). However, with residual HXK1 protein detected in *C. zofin-giensis hxk1* strains, we cannot exclude that "strain-independent" +Glc responses acted through HXK1, but can be activated at the lower HXK1 concentrations found in the mutants. Because we were unable to delineate strain-independent +Glc responses from low-activation HXK1 responses, we did not explicitly aim to elucidate the mechanism of HXK1 in Glc-signaling with this dataset but rather use the lack of photosynthetic repression or TAG-accumulation in *hxk1* strains as controls for gene discovery.

## Combined glucose and iron effects on the photosynthetic electron transport chain

Photosynthetic rates, pigments, and protein abundances showed major changes in the photosynthetic ETC across conditions. Both $F_v/F_m$ and oxygen evolution measurements showed that only WT−Fe +Glc cells were non-photosynthetic at 84 h (Fig. 4a–d; Supplementary Fig. 6d, Supplementary Data 4). Concurrently, the mean difference in abundance of all photosystem subunits was ~35-fold between highest (+Fe−Glc, all strains) and lowest (WT−Fe+Glc) conditions, but iron-rich PSI subunits had a mean abundance range of ~70-fold (Fig. 5a).

Due to observed differences in pigmentation across treatments (Supplementary Fig. 6a), we conducted HPLC analysis of chlorophylls and carotenoids. Astaxanthin and zeaxanthin accumulated only in WT +Glc cells (Supplementary Fig. 7). Normalized per cell, +Fe significantly increased chlorophyll levels compared to all −Fe samples. In addition, WT−Fe+Glc had the lowest total chlorophyll per cell of any condition (Fig. 4e), consistent with previous results[12,13]. Due to the large increase in the volume of WT+Fe+Glc cells, likely from the buildup of starch and lipids (Fig. 2, Supplementary Fig. 2a, c), total chlorophyll on a per biomass basis was equally low between WT−Fe+Glc and WT+Fe+Glc (Fig. 4f). However, WT+Fe+Glc had the highest chlorophyll *a/b* ratio of all conditions (Supplementary Fig. 6b) indicating a lower ratio of light-harvesting chlorophyll-binding proteins (which contain chlorophyll *b*) to photosynthetic reaction centers which evolve oxygen and contain chlorophyll *a*. The high chlorophyll *a/b* ratios may be due to +Fe preferentially upregulating reaction centers (see Fig. 6a), which contain iron cofactors or +Glc preferentially downregulating periphery complexes. Consistent with reaction center maintenance, gross oxygen evolution normalized by chlorophyll was ~3-fold higher in WT+Fe +Glc than any other photosynthetic condition (Supplementary Fig. 6c). In total, while there may be less chlorophyll per biomass in WT+Fe+Glc cells compared to autotrophic conditions, the remaining chlorophyll in WT+Fe+Glc is prioritized for oxygen evolution.

## Proteomics signatures of heterotrophic cultures reveal conserved players of photosynthesis regulation

Because only the WT−Fe+Glc cells switched off photosynthesis, we hypothesized that proteins that were lowest in the heterotrophic state ("lowest-in-heterotrophy") would include essential photosynthetic players. Conversely, "highest-in-heterotrophy" proteins would likely include enzymes that maintain heterotrophic metabolism in the absence of photosynthesis. A statistical enrichment test (see Methods for details) identified 274 lowest-in-heterotrophy and 305 highest-in-heterotrophy proteins. These proteins passed a differential abundance ($>\log_2 = 0.5$) and statistical significance threshold (adjusted *p* value) for being uniquely down- or upregulated in WT−Fe+Glc vs. all other conditions that maintained photosynthesis (Fig. 5b, c). Lowest/highest-in-heterotrophy proteins represented various linear model subgroup classifications, showing that these proteins could individually have distinct Fe or Glc responsiveness across all other conditions (Supplementary Fig. 8).

Observing an attenuated −Fe+Glc repression/induction of protein levels in *hxk1* strains, we expanded the list of lowest/highest-in-heterotrophy proteins. For example, ~70% of the statistically stringent lowest-in-heterotrophy proteins had second lowest abundance in

either the *hxk1-2* − Fe+Glc or *hxk1-1* − Fe+Glc conditions. Because HXK1 was detectable at low levels in both mutant strains, residual HXK1 activity may have resulted in attenuated photosynthetic protein repression in *hxk1* strains. Therefore, we included proteins that were lowest or highest in −Fe+Glc conditions regardless of strain, leading to expanded lowest- and highest-in-heterotrophy lists of 1433 and 1088 proteins, respectively (Supplementary Data 5).

Known photosynthesis proteins were well represented among the 1433 lowest-in-heterotrophy proteins (Fig. 5b, c). This included 38/52 (~69%) of the detected photosystem proteins, eight Calvin-Benson cycle enzymes (e.g., RBCS1, RCA2, FBP2), 14 chlorophyll/heme biosynthetic enzymes (e.g., POR1, CHLG1, ChlB), and nine photosystem assembly factors (e.g., PPD3/4, HCF136, OXA1, LPA1B). Consistently, GO terms related to photosynthesis (GO:0009768, GO:0016168) were also enriched (Fisher Exact Test, $p = 0.00047, 0.0005$, Supplementary Data 5).

Most lowest-in-heterotrophy proteins do not appear to have been genetically characterized in any organism. However, 767 (64%) nucleus-encoded lowest-in-heterotrophy proteins have at least one predicted ortholog in a vascular plant database, while 1083 (90%) proteins have a *C. reinhardtii* ortholog. We developed a pipeline to predict if putative orthologous groups between *C. reinhardtii* and *C. zofingiensis* had significantly conserved photosynthetic associations (Methods, Supplementary Fig. 9a, Supplementary Data 5). The *C. reinhardtii* orthologs of lowest-in-heterotrophy proteins from *C. zofingiensis* (1060 unique ortholog groups) had significant overlap with GreenCut2 (113 out of 447 total GreenCut2 matched ortholog groups, Fisher Exact Test, $p = 1.2 \times 10^{-8}$)[42], and especially genes in mRNA co-expression networks (photosynthesis network: 217/812 groups, $p = 2.2 \times 10^{-19}$; tetrapyrrole network: 165/608 groups, $p = 2.7 \times 10^{-15}$)[43] (Supplementary Fig. 9a, Supplementary Data 5). Shared orthologs to *C. zofingiensis* lowest-in-heterotrophy proteins were also found in *C. reinhardtii* photosynthetic mutant libraries, but with lower overlap significance (Acetate-requiring[7]: 33/165 groups, $p = 0.10$; CLIP photosynthesis-defective[8]: 32/161 groups, $p = 0.12$). Overall, 315 lowest-in-heterotrophy *C. zofingiensis* proteins were identified as conserved green algal proteins whose orthologs were independently associated with photosynthesis across *C. reinhardtii* studies (Supplementary Data 5).

We analyzed photosynthesis gene expression across the larger evolutionary span between green algae and vascular plants. First, we compared our dataset with a proteomic time course of *A. thaliana* de-etiolation, during which photosynthesis develops after heterotrophic seedlings perceive light[34]. Differentially upregulated proteins for photosynthetically mature cotyledons (96 h, >2-fold change) had highly significant ortholog overlap with *C. zofingiensis* lowest-in-heterotrophy groups (173 out of 372 *C. zofingiensis* lowest-in-heterotrophy and 643 *A. thaliana* upregulated ortholog members, $p = 3.1 \times 10^{-19}$, Supplementary Fig. 9b). We focused on 97 nucleus-encoded orthologous groups sharing a one-to-one copy number relationship, which strongly indicates conserved ortholog function. Along with known photosynthesis proteins (e.g., LHCs, chlorophyll biosynthetic enzymes) were several proteins lacking annotation[44] related to photosynthesis or chlorophyll (56 orthologs) or without mutant validation of biological function (59 orthologs, Supplementary Data 5, Fig. 5). Uncharacterized genes include five embryo-defective (*EMB*) *A. thaliana* genes, whose photosynthetic function would likely not have been revealed by forward genetics. Interestingly, the conserved orthologs included multifunctional NEET proteins, which play a role in 2Fe-2S cluster biosynthesis in chloroplasts and mitochondria[45]. This result implies a conserved coupling of iron cofactor assembly to photosystem assembly across plants and green algae.

We additionally compared *C. zofingiensis* lowest-in-heterotrophy proteins to their maize (*Zea mays*) and rice (*Oryza sativa*) orthologs. In monocots, transcriptional studies have tracked photosynthesis as it

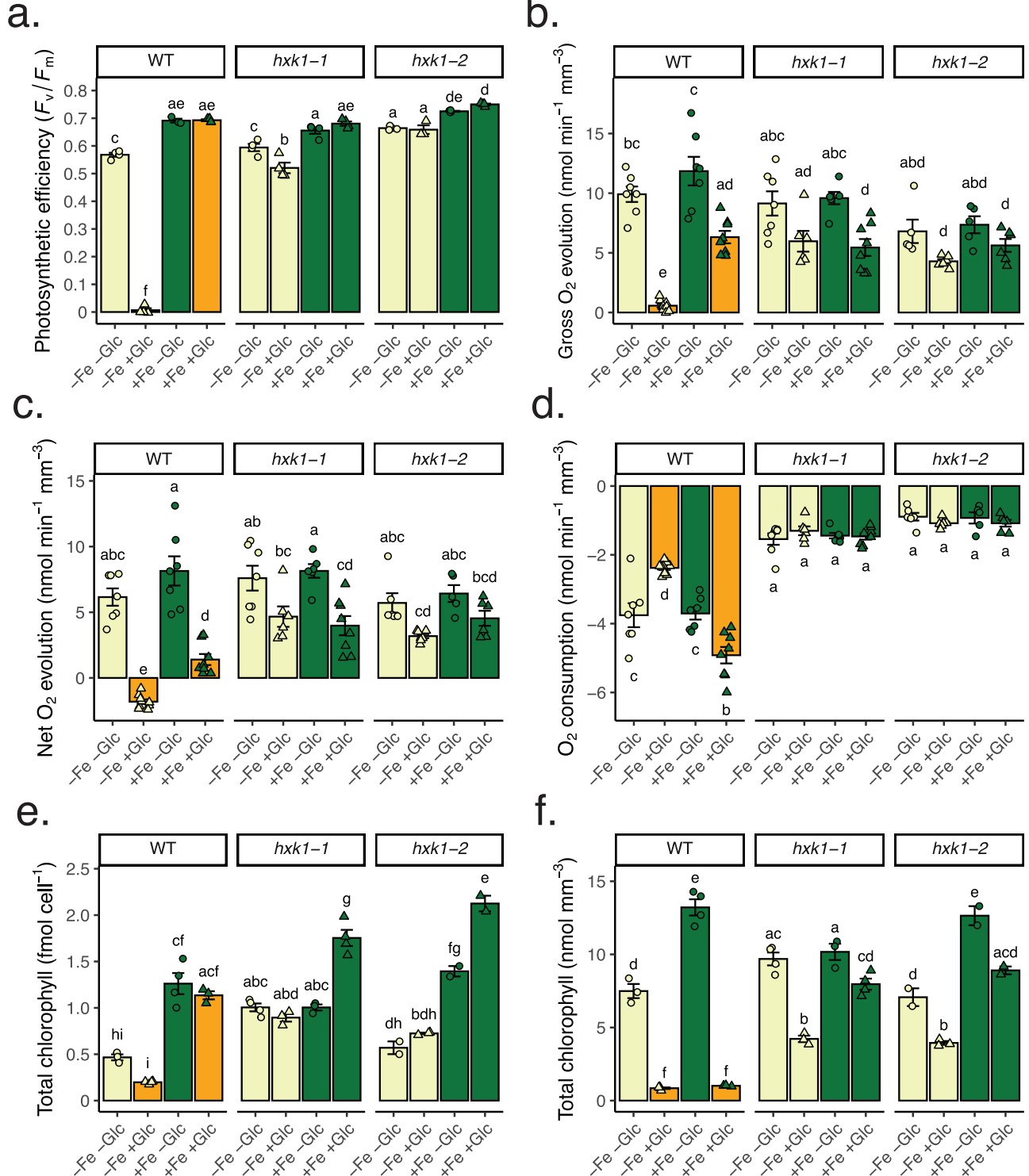

**Fig. 4 | Photosynthetic and chlorophyll measurements of iron and glucose treatments across strains. a** Photosynthetic efficiency of PSII ($F_v/F_m$) of 12 conditions at 84 h, same cultures used for proteomics. Data represent means ± SE ($n = 3$–4, individual data points shown, one experimental batch). $F_v/F_m$ measurements are from the same samples used for proteomics. **b** Gross oxygen evolution (net oxygen evolution light minus oxygen uptake rate in the dark, normalized per biomass). **c** Net oxygen evolution in the light. **d** Oxygen consumption in the dark.

All oxygen rate data (**b**–**d**) represent means + SE of combined replicates of two experimental batches, $n = 6$–8. **e** Total chlorophyll ($a + b$ in femtomoles per cell) at 84 h. **f** Total chlorophyll ($a + b$ in femtomoles per biomass) at 84 h. Chlorophyll data (**e**, **f**) represent means ± SE ($n = 2$–4, one experimental batch). Statistical comparisons are applied by Tukey's HSD with adjusted $p = 0.05$ and represented by letters above bars. Source data are provided in the Source Data File.

develops across the basipetal leaf axis[46]. Lowest-in-heterotrophy proteins had significant overlap with orthologs upregulated in maturing photosynthetic tissue, both in *Z. mays* (up to 201 out of 670 *C. zofingiensis* and 769 *Z. mays* members, $p = 5.8 \times 10^{-17}$, Supplementary

Fig. 9c) and *O. sativa* (up to 135 out of 667 in *C. zofingiensis* and 461 in *O. sativa*, $p = 3.2 \times 10^{-15}$, Supplementary Fig. 9d).

In total, 154 *C. zofingiensis* lowest-in-heterotrophy proteins had orthologs also associated with photosynthesis in at least 3 of the 4

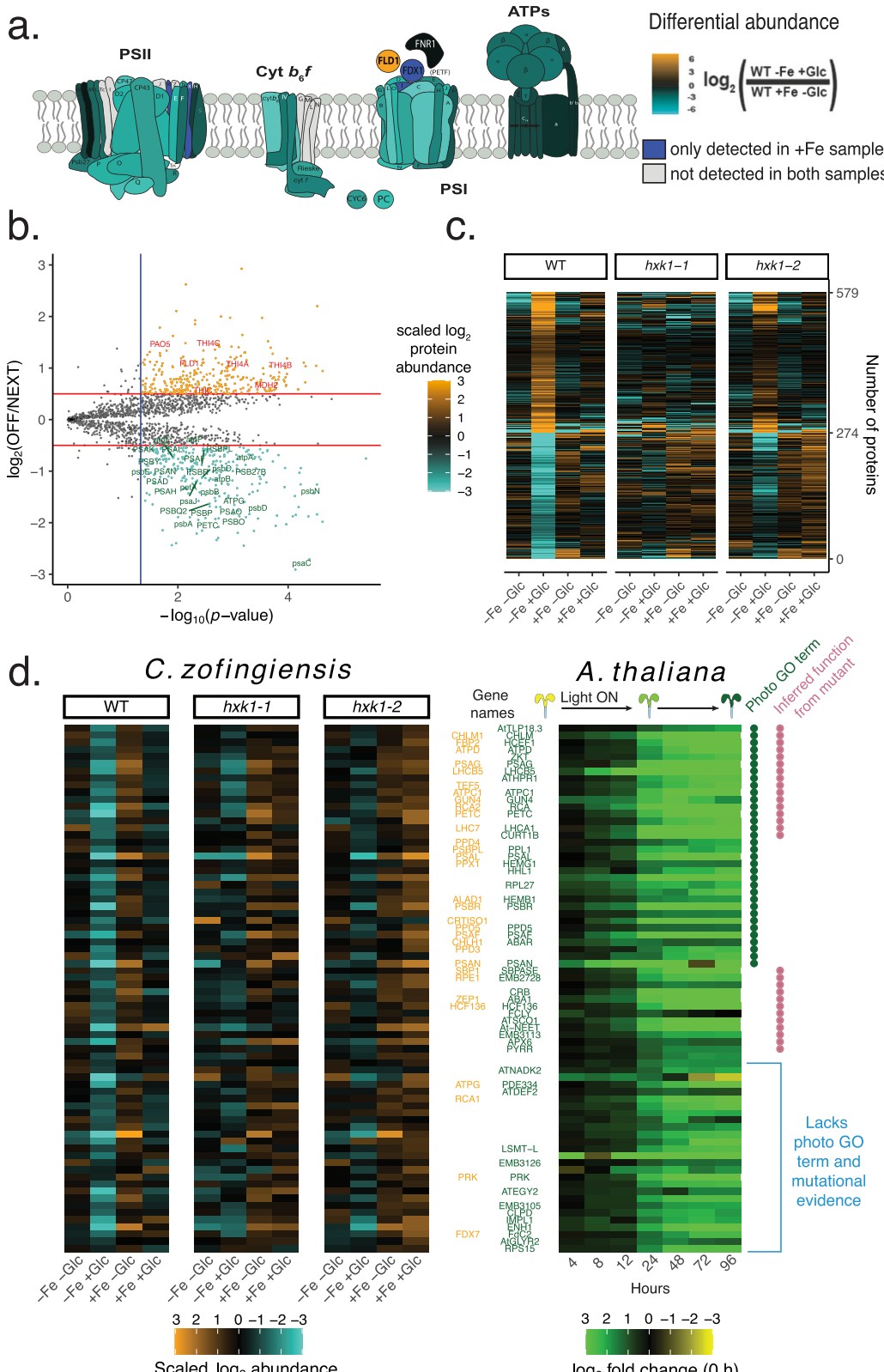

comparison species (*C. reinhardtii*, *A. thaliana*, *Z. mays*, and *O. sativa*, Supplementary Data 5). Orthologs with this high evolutionary frequency association with photosynthesis notably included several amino acid related enzymes (e.g., glutamate synthase, shikimate kinases, and alanine aminotransferase), implying a conserved regulatory link between nitrogen metabolism and photosystem maturation across diverse photosynthetic organisms.

Highest-in-heterotrophy proteins included likely repressors of photosynthesis or mediators of heterotrophic metabolism. For example, pheophorbide *a* oxygenase 5 (PAO5), a chlorophyll degradation enzyme containing a 2Fe-2S cluster, could function in depletion of chlorophyll during the transition to heterotrophy. Malate dehydrogenase 2 (MDH2) could be involved in shuttling peroxisomal reductant to the chloroplast[47], where non-photosynthetic sources of

**Fig. 5 | Proteomic enrichment of photosynthetic *C. zofingiensis* proteins are evolutionarily conserved. a** Diagram heatmap of photosystem protein subunit fold change differences in log$_2$(WT−Fe+Glc/WT+Fe−Glc). **b** Volcano plot results of statistical pipeline to isolate proteins that were significantly highest or lowest in the heterotrophy (WT−Fe+Glc). Proteins that are stringently highest-(orange) or lowest-in-heterotrophy (turquoise) are in color. They pass a 5% FDR-adjusted *p* value cutoff (blue line) and a 0.5 fold change cutoff (red line). Select proteins annotations that either represent photosystems (green text) or highest-in-heterotrophy proteins (red) described in the text. The *y*-axis value is the log$_2$ fold change of the highest (positive) or lowest (negative) in heterotrophy to the next highest/lowest photosynthetic condition. The *x*-axis shows the −log$_{10}$ maximum *p* value of all highest/lowest WT−Fe+Glc comparisons from any other condition. **c** Heatmap of the highly significant highest-and lowest-in-heterotrophy proteins across all experiments. **d** Heatmaps show the protein abundance of single copy and nuclear-encoded orthologs between *C. zofingiensis* (left three heatmaps) and *A. thaliana* (right) where each ortholog is associated with photosynthesis in both

species. Rows represent orthologous relationships, where a *C. zofingiensis* and *A. thaliana* proteins in the same row are one-to-one orthologs of each other. The *A. thaliana* heatmap is plotted with all mean time point abundances log$_2$ normalized to the 0 h timepoint of etiolated cotyledons. Data for *A. thaliana* are from a de-etiolation experiment (ref. 34) represented by the infographic at the top of the heatmap. All included proteins had log$_2$ fold change >1 at 96. Where available, manually annotated abbreviations from *C. zofingiensis*[11] are labeled in the middle in orange and Araport11 aliases are labeled in green. The *A. thaliana* GO slim database (updated 01-01-2021)[43] was used to extract which genes had a photosynthetic light harvesting associated GO term (green dot, column on right) and genes that were "inferred from mutant phenotype" (red dot GO evidence code: "IMP") are also labeled. The bottom set of orthologs lacks both photosynthesis GO and mutational evidence in any condition according to GO slim database. PSI/II (photosystem I/II), Cyt $b_6f$ (cytochrome $b_6f$), ATPs (plastid ATP synthase), FLD1 (flavodoxin 1), FNR1 (flavodoxin-NADP reductase 1), FDX1 (ferredoxin 1 or PETF). Source data are provided in the Source Data File.

NADPH would be necessary for *dn*FAS and other metabolism in heterotrophic plastids. In fact, many highest-in-heterotrophy proteins contain predicted oxidoreductase or dehydrogenase domains (e.g., Cz05g05160, Cz12g08230, Cz03g32060), and their induction could support the redox needs of cells that lack photosynthetic reductant. Highest-in-heterotrophy proteins also included enzymes involved in sulfur and thiazole metabolism (Supplementary Information, Supplementary Fig. 10). The upregulation of iron-rich enzymes in thiazole synthesis (THIC, THI1/4) and PAO5 is notable, because these enzymes may represent iron sinks in heterotrophy.

## Mitochondrial iron proteins are prioritized over plastid proteins in heterotrophy

By comparing respiratory vs. photosynthetic ETC proteins and the likely orthologs of mitochondrial vs. plastid Fe importers, we found that Fe resources were prioritized to mitochondria in heterotrophic *C. zofingiensis* cells. Photosynthetic ETC subunits decreased greatly in −Fe compared to the respiratory ETC across all strains and Glc states (Fig. 6a). Specifically in WT, the mean photosynthetic ETC vs. respiratory ETC abundance ratio (log$_2$ (−Fe$_{abundance}$/+Fe$_{abundance}$)) was −1.6 vs. −0.3 in −Glc (Wilcoxon Rank Sum Test, $W = 1284$, $p = 3.3 \times 10^{-10}$, Supplementary Data 6) and −2.0 vs. −0.6 in +Glc (Wilcoxon, $W = 1367$, $p = 2.0 \times 10^{-9}$). The photosynthetic vs. respiratory ETC difference in −Fe was especially striking for proteins that bind iron cofactors (Fig. 6a, red points). Within the photosynthetic ETC, iron-rich photosynthetic complexes declined more in −Fe (PSI, mean log$_2$(−Fe/+Fe) = −2.48; Cyt $b_6f$, mean log$_2$ = −2.19) than PSII (mean log$_2$ = −1.47) and chloroplast ATP synthase (mean log$_2$ = −0.978). Altogether, iron deficiency depletes photosynthesis while weakly impacting respiration in *C. zofingiensis*.

Differences in transporter protein levels in *C. zofingiensis* suggest possible mechanisms of iron trafficking that prioritize respiration. First, −Fe conditions resulted in a clear increase in high-affinity Fe assimilation and import proteins at the plasma membrane (FEA1, FEA2, FOX1, FRE1, Fig. 6b)[22,37]. Additionally, MIT1/MFL1, a mitoferrin-family protein whose orthologs in plants and yeast import Fe into mitochondria[48–50], was upregulated in both −Fe+Glc and +Fe+Glc (Fig. 6b). Conversely, orthologs of putative plastid Fe transporters were downregulated in heterotrophy, with TIC21B[51,52] being down-regulated by −Fe, ABCI10 by +Glc in WT and *hxk1-2*, and ABCI12[53] by −Fe+Glc (Fig. 6b). Taken together, changes in transporter proteins suggests that limited Fe is preferentially directed to the mitochondria during heterotrophy.

## TAG accumulates from de novo fatty acid biosynthesis regardless of photosynthetic state

Because mixotrophic cells (WT+Fe+Glc) accumulated TAG but maintained photosynthesis, whereas heterotrophic cells (WT−Fe+Glc)

accumulated TAG but lost photosynthesis (Figs. 4, 7a), we were able to distinguish proteins associated with Glc-mediated photosynthesis repression from the proteins supporting TAG accumulation. For example, major lipid droplet protein 1 (MLDP1), a biomarker of green algal lipid droplets[54,55], was -8-fold higher in both WT+Fe+Glc and WT−Fe+Glc than any other condition (Fig. 7b). The same statistical thresholds used to find highest/lowest-in-heterotrophy proteins were applied to find "highest-during-TAG" and "lowest-during-TAG" proteins in both WT+Glc conditions. We prioritized 112 highest- and 39 lowest-during-TAG proteins passing a log$_2$ = 0.5 and 5% FDR cutoff (Fig. 7c, d; expanded less-stringent lists are in Supplementary Data 7). Highest-during-TAG proteins were enriched for the "fatty acid biosynthetic process" GO term (GO:0006633, $p = 0.0026$). Significant overlap of 37 highest-during-TAG proteins (-9% total) with lipid droplet fractions isolated from *C. zofingiensis* cells grown under photoautotrophic nitrogen deprivation (Fisher Exact Test, $p = 2.2 \times 10^{-16}$, Supplementary Data 7)[55] included MLDP1 and a caleosin-related protein (CAS1, Cz09g31050).

Highest-during-TAG proteins represented a nearly complete pathway of plastid-localized *dn*FAS and ER-localized TAG condensation (Fig. 7e). In addition, high levels of four subunits of the plastid pyruvate dehydrogenase complex (PDH) and seven glycolytic enzymes (PFK2, IPGM1, PGM2&5, PGH1, PYK1&5) likely revealed the dominant route of pyruvate conversion to acetyl-CoA as a substrate for *dn*FAS.

Highest-during-TAG proteins corroborated previous transcriptome data showing that *dn*FAS upregulation is an underlying mechanism of TAG accumulation in *C. zofingiensis*[13]. We mined published transcriptome data of *C. zofingiensis* for the *dn*FAS pathway and compared it to the orthologous pathway in *C. reinhardtii* (Supplementary Fig. 11)[10,12,35]. Either in response to +Glc or photoautotrophic nitrogen deprivation, the entire *dn*FAS pathway is highly and rapidly upregulated in *C. zofingiensis*[10,12], whereas the orthologs in *C. reinhardtii* are downregulated during nitrogen deprivation[35]. *dn*FAS downregulation is consistent with research showing that *C. reinhardtii* accumulates TAG by remobilizing membrane lipids rather than increasing *dn*FAS, while both processes may drive *C. zofingiensis* TAG accumulation[9,14]. Therefore, we confirmed with proteomics that the rapid molecular upregulation of *dn*FAS is a distinct *C. zofingiensis* strategy to accumulate TAGs.

## A group of ferredoxins and desaturases support TAG accumulation despite Fe deficiency

FADs and FDXs are often impacted by Fe deficiency in *C. reinhardtii*[32,56] and marine phytoplankton[25,37,39]. However, some FADs and FDXs were highest-during-TAG proteins, meaning they were strongly upregulated despite iron deficiency in the WT−Fe+Glc condition (Fig. 8a). While FLD1, which replaces FDX activity, was upregulated in all −Fe conditions (Fig. 8b), two FDX domain

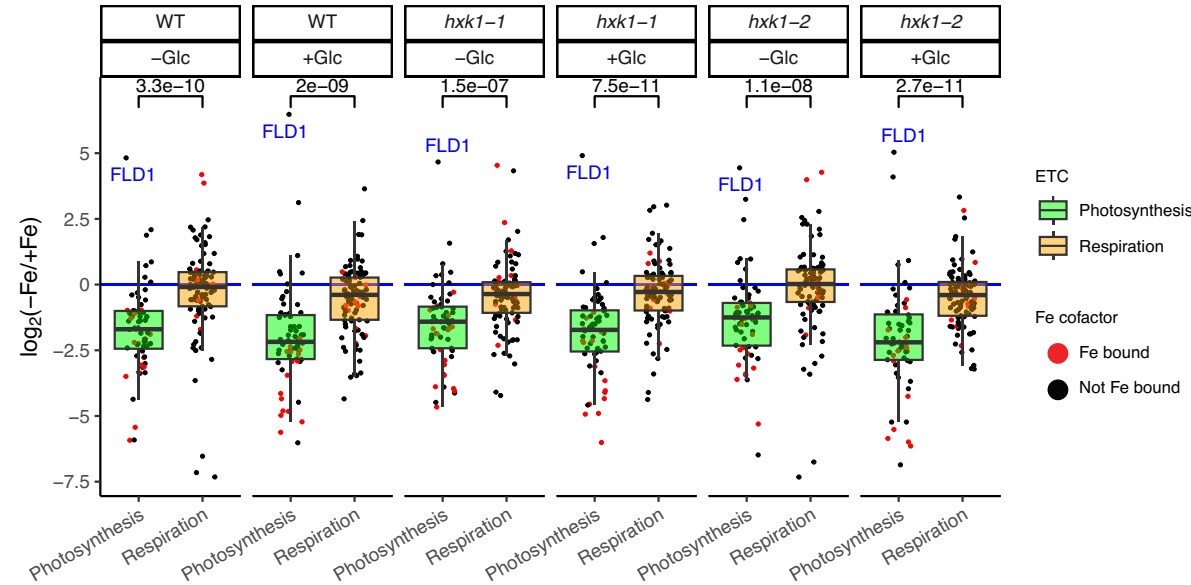

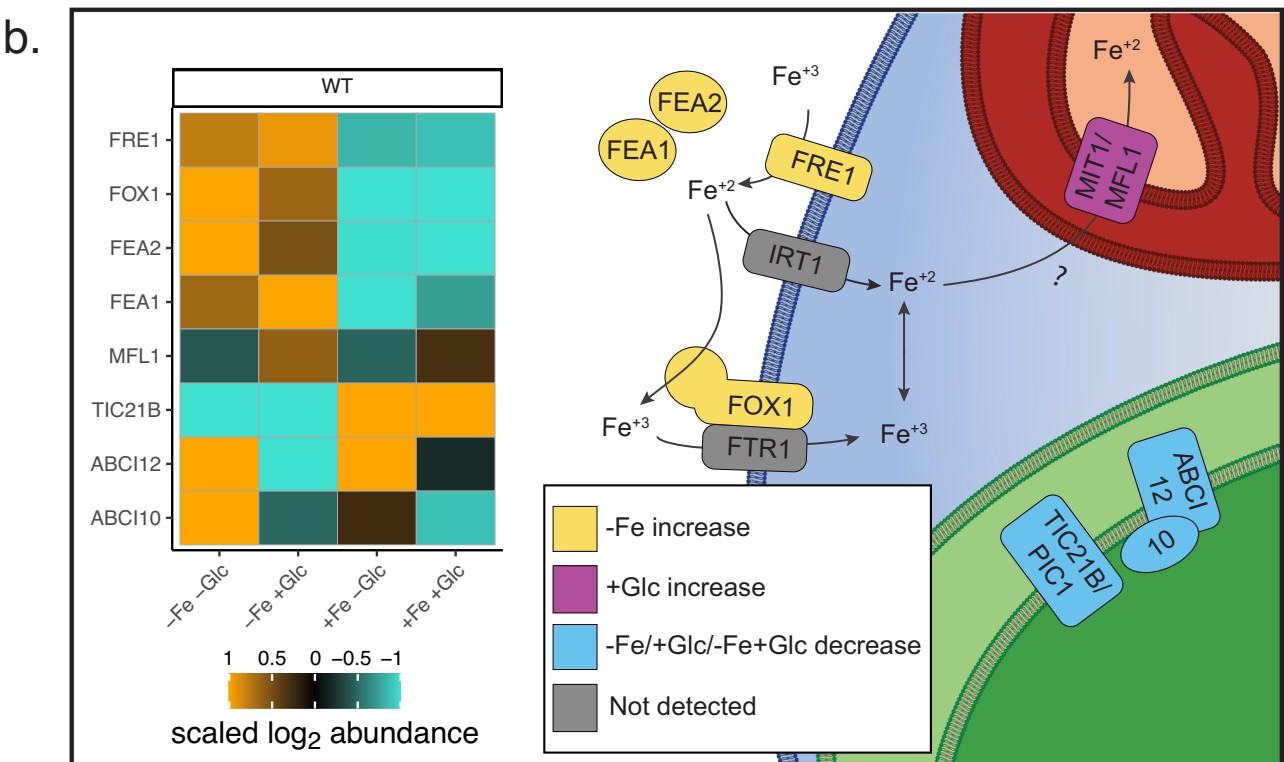

**Fig. 6 | Iron-rich mitochondrial processes are preferentially maintained over plastid processes. a** Box plot of mean $\log_2$(−Fe/+Fe) abundance of protein subunits in the photosynthesis ETC (*x*-axis, "Photosynthesis", *n* = 64 proteins) vs. the respiratory ETC ("Respiration", *n* = 97 proteins) in every Glc and strain combination. The box represents first to third quartile (Q1 & Q3) with median as center line, while the whiskers represent local minimum (Q1 X 1.5 X interquartile range) and maximum (Q3 X 1.5 X interquartile range). Individual points for each subunit are shown. Red points are subunits known to bind to an iron cofactor (heme, Fe-S cluster, non-heme iron). The *p* value for the Wilcoxon Rank Sum Test comparing Fe fold change

difference of Photosynthesis vs. Respiration is shown for each Glc and strain combination. The Photosynthesis outlier FLD1 (flavodoxin 1) is labeled which is upregulated in all −Fe conditions. **b** Protein heatmap (left) and model of iron-trafficking components in *C. zofingiensis* (right). The high-affinity iron assimilation pathway components were highly upregulated in −Fe and localize to the plasma membrane or periplasm[37]. The transporter orthologs IRT1 and FTR1 occur in the *C. zofingiensis* genome but were not detected in enough conditions to make interpretations on their regulation. Source data are provided in the Source Data File.

proteins (FDX2, FDX5) were upregulated in WT−Fe+Glc *and* WT+Fe+Glc (Fig. 8a). Other FDX proteins, including PETF/FDX1, which drives photosynthetic NADPH reduction, were depleted by −Fe or −Fe+Glc (Fig. 8c).

Di-iron-containing fatty acid desaturases SAD1, FAD2, and FAD7A were also among the highest-during-TAG proteins (Fig. 8d). FADs in *C. reinhardtii* are repressed during iron deficiency, particularly the ortholog of CzSAD1, CrFAB1[56]. Besides the three upregulated

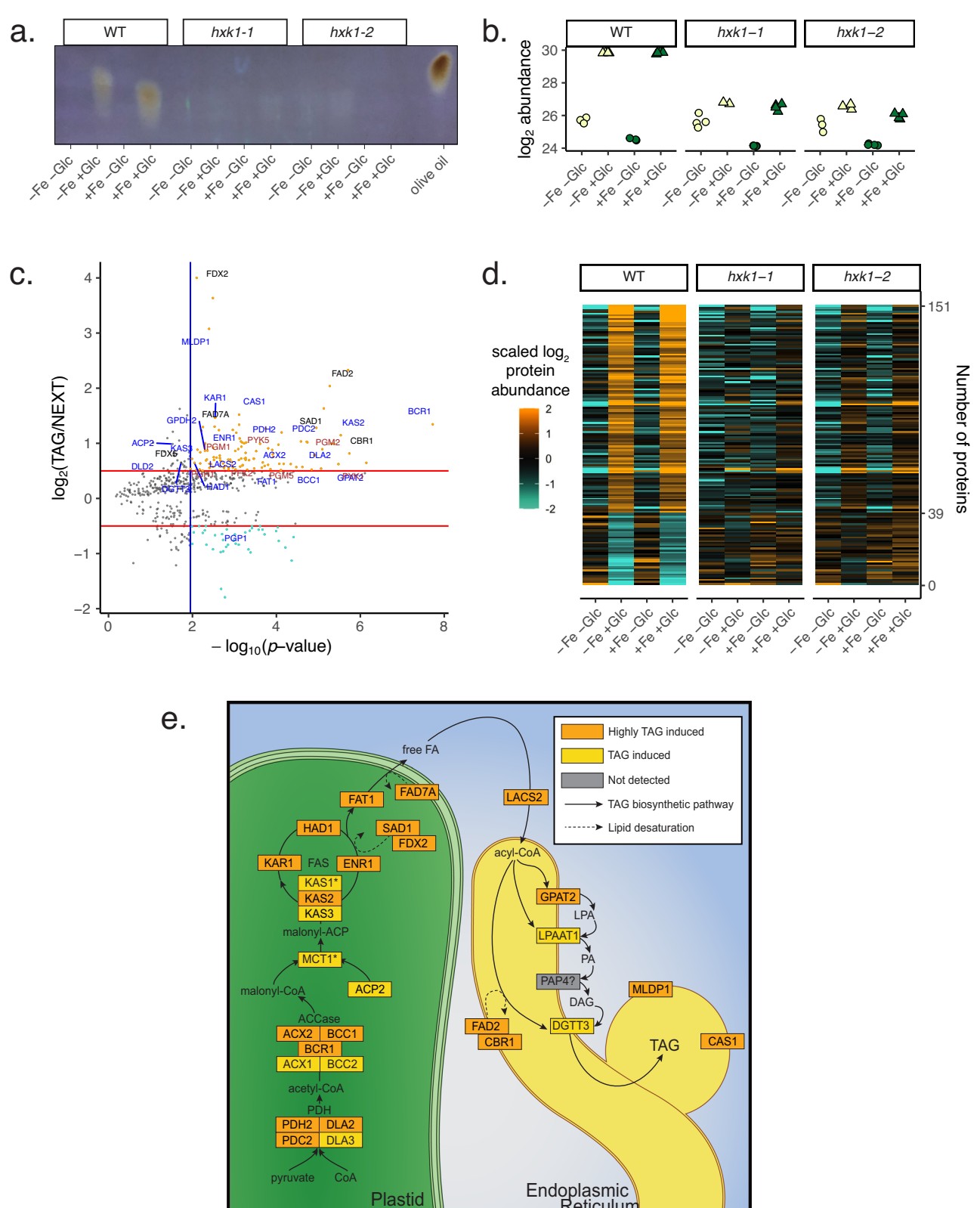

desaturases in *C. zofingiensis*, most FADs were lowest in −Fe or −Fe+Glc (Fig. 8e, Supplementary Data 8). By integrating cross-species comparisons and previous biochemical assays[57,58], we propose that FDXs and FADs maintained in WT−Fe+Glc are functionally preferred for *dn*FAS-based TAG accumulation, whereas the FADs that are decreased in −Fe are more likely to play a role in thylakoid membrane synthesis (see Discussion). Therefore, our proteomics revealed that

photosynthesis loss in WT−Fe+Glc in *C. zofingiensis* was not only due to prioritizing iron for respiratory ETC over photosynthetic ETC, but also TAG accumulation over thylakoid biogenesis.

## Discussion
In this study, we found that lack of sufficient iron is a key requirement for *C. zofingiensis* to switch off photosynthesis in the light when grown

**Fig. 7 | Concerted upregulation of de novo fatty acid synthesis in both WT+Glc proteomes. a** UV-TLC showing high TAG accumulation in WT−Fe+Glc and WT+Fe+Glc at 84 h (equal volumetric biomass loaded with olive oil TAG standard). **b** Log$_2$ abundance of the major lipid droplet protein 1 (MLDP1) with Glc and Fe at 84 h **c** Volcano plot results of statistical pipeline to isolate proteins that were significantly highest or lowest in both TAG-accumulating conditions at 84 h (WT−Fe+Glc, WT+Fe+Glc). Proteins that pass statistical threshold are in color with highest (orange) and lowest in TAG (turquoise). The labeled proteins include the manually annotated lipid biosynthesis pathway (blue), glycolysis (brown), and proteins putatively involved in fatty acid desaturation (black, FDX2, CBR1, FAD2, FAD7A). All other axes and plot features are the same as Fig. 4e. **d** Heatmap of proteins associated with TAG accumulation that were highest-during-TAG (112 proteins) or

lowest-during-TAG (39 proteins) across all proteomes and pass significance threshold. **e** Schematic of induced enzymes that likely play a role in the TAG accumulation mechanism, showing a concerted upregulation of de novo fatty acid synthesis (plastid enzymes), with a complete pathway induced. A subset of proteins involved in TAG condensation were also upregulated. PAP4 was not detected in this study, but its induction has been noted at the transcriptional level in previous studies[10,12]. Proteins pass statistical significance from volcano are labeled orange. Most orange-yellow proteins were additionally highest in WT+Glc conditions but did not exceed the fold change stringency cut-off. MCT1 and KAS1 (with *) were induced by +Glc in WT but were also induced in one or both *hxk1*+Glc strains. Source data are provided in the Source Data File.

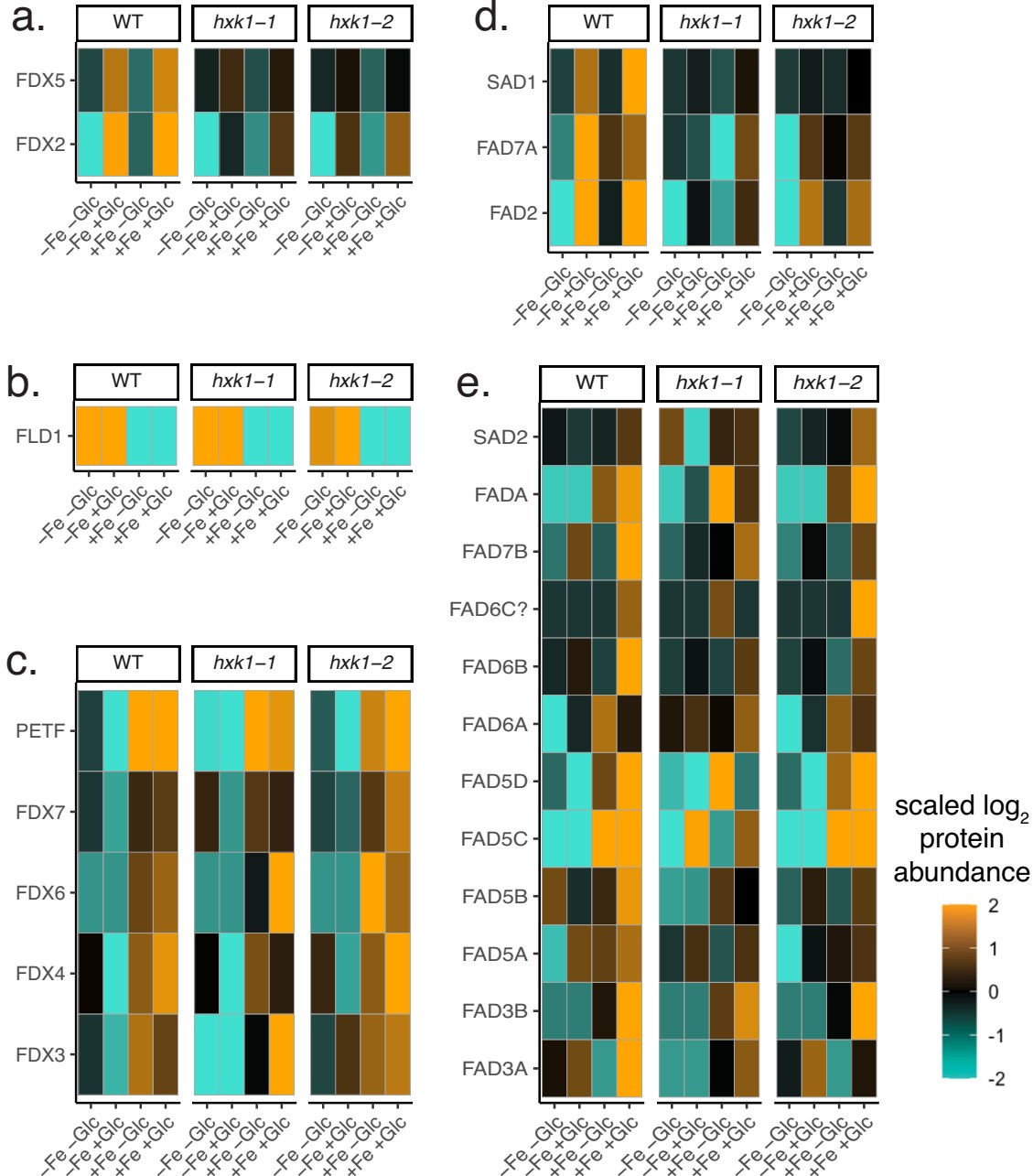

**Fig. 8 | Differential prioritization of ferredoxins and fatty acid desaturases in TAG-accumulating conditions. a** Heatmap of two ferredoxins (FDX) highly enriched for TAG accumulation in WT+Glc (highest-during-TAG, also see Fig. 7c). **b** Heatmap of flavodoxin 1 (FLD1, −Fe induced). **c** Remaining annotated *C. zofingiensis* ferredoxins. **d** Heatmap of three highest-during-TAG fatty acid desaturases

(FAD or SAD, stearoyl ACP desaturase). **e** Remaining annotated the desaturases were repressed by −Fe or −Fe+Glc, but many were highly induced in WT+Fe+Glc. All heatmap ranges are scaled by the same scale color bar. Source data are provided in the Source Data File.

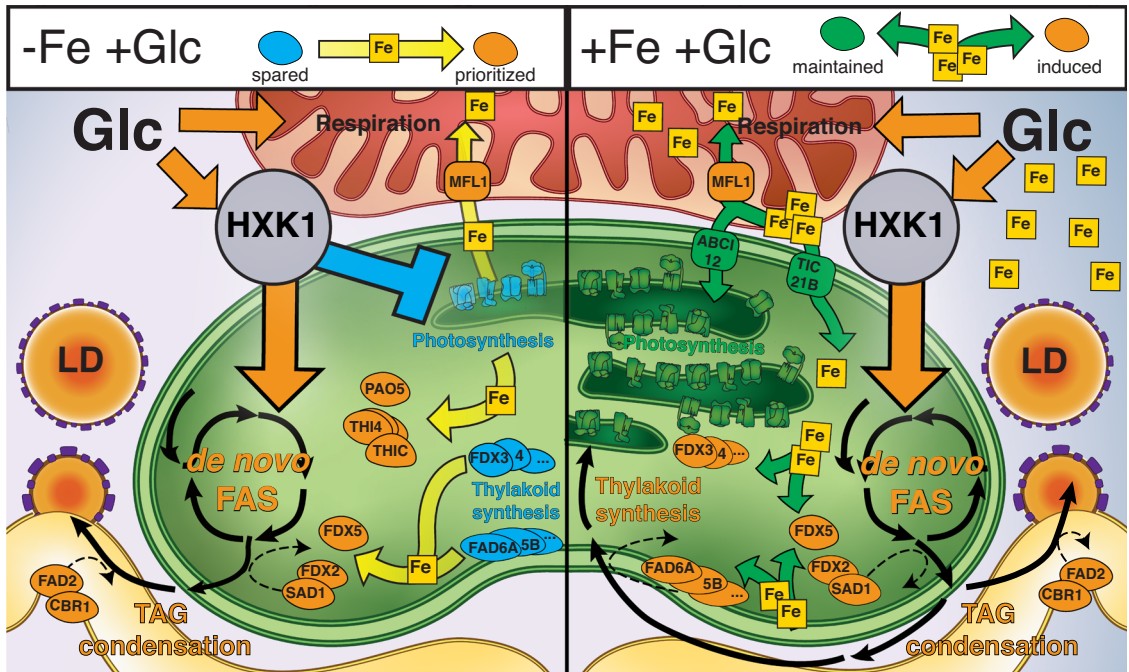

**Fig. 9 | Overview of intracellular iron partitioning strategies behind −Fe's role in heterotrophy and TAG accumulation.** The panel on the left summarizes predicted intracellular iron prioritizations in WT−Fe+Glc that underly the glucose-mediated switch-off of oxygenic photosynthesis during TAG accumulation. Yellow arrows indicate a potential iron-prioritization from a repressed iron-containing proteins (turquoise) that are spared when a HXK1-mediated GLC signaling pathway activates respiration and *dn*FAS. Prioritized and induced iron-containing proteins are labeled (orange). When photosynthetic ETC is repressed, the released iron is potentially trafficked from the plastid to the mitochondria via MFL1 which is upregulated in +Glc. Additionally, spared iron proteins likely provide Fe resources for the upregulated *dn*FAS pathway, including iron-rich players conducting redox transfers of lipid desaturation: FDX2 to SAD1 and CBR1 to FAD2 (See Discussion). These proteins exhibit upregulation in TAG even though most ferredoxins and desaturases are depleted, suggesting their specific prioritization to support *dn*FAS during TAG accumulation. Thylakoid membranes are likely depleted in WT−Fe+Glc due to loss of membrane synthesis supported by most desaturases. Fe may also be reallocated to iron-containing proteins THIC, THI4A-C, and PAO5 which are uniquely upregulated in the heterotrophic WT−Fe+Glc conditions. In the right panel, photosynthesis is maintained during TAG accumulation due to replete Fe. The green arrows represent how replete iron in WT+Fe+Glc allows enough iron resources to be distributed to the pathways that would have been depleted in iron deficiency. While +Glc still triggers a HXK1-dependent upregulation (orange) of *dn*FAS, +Fe is distributed to both TAG-supporting and thylakoid supporting desaturases, the latter of which are highly induced in the mixotrophic WT+Fe+Glc condition. Both thylakoids and photosynthetic complexes are maintained due to replete iron (green=maintained). While MFL1 is still upregulated by +Glc, putative thylakoid importers (TIC21B, ABCI12) are maintained. Lipid droplets (LD) accumulate in both conditions through the +Glc-HXK1 pathway. The organelles in the diagram are plastid (green), mitochondria (red), cytoplasm, (blue), and endoplasmic reticulum (yellow). Full names of protein abbreviations can be found in Supplementary Data 9.

on Glc. High-resolution quantitative proteomics showed that heterotrophic cells prioritize respiration and TAG accumulation over photosynthesis and thylakoid biosynthesis while increasing biomass. Iron-containing proteins that are uniquely upregulated in WT−Fe+Glc (PAO5, THI1/4, THIC) may also further divert iron resources from photosynthesis to support heterotrophy. In Fig. 9, we summarize the roles of iron prioritization in mediating the loss of photosynthesis in WT−Fe+Glc vs. WT+Fe+Glc during TAG accumulation. We provide evidence that replete iron rescues photosynthesis despite HXK1-mediated Glc signaling in WT+Fe+Glc, because there is sufficient iron to simultaneously support the iron-cofactor proteins of photosynthesis, respiration, thylakoid biosynthesis, and TAG accumulation.

Our full-factorial experimental design revealed Glc-responsive pathways that are also dependent on iron nutritional status, which provided insights into evolutionarily distinct iron demands in *C. zofingiensis*. Recent studies in plants show molecular networks respond more to combinatorial signals than to individual signals. For example, more rice genes respond to nitrogen and water interactions than to these inputs alone[59]. In plants, iron deficiency networks integrate hormonal and other nutrient responses[60], but to our knowledge the impact of sugar feedback on −Fe responses has not been investigated. As energetic and biosynthetic pathways are responsive to substrate (e.g., Glc) and cofactor demands (e.g., iron), synergistic iron vs. sugar studies may reveal evolutionary differences in pathway prioritizations across photosynthetic species.

Because *C. zofingiensis* accumulates TAGs via *dn*FAS concurrently with an increase in biomass, it is a powerful reference organism that can inform on how to engineer algal biofuels without inhibiting growth. We identified candidate proteins involved in TAG accumulation by distinguishing lipid accumulation from photosynthetic repression and forcing iron prioritization of the most essential iron-cofactor proteins supporting *dn*FAS. For example, the subset of FDXs and FADs that are upregulated in WT−Fe+Glc when *dn*FAS is upregulated are promising genetic targets to improve biofuel production. FDXs being upregulated in a −Fe condition suggests FLD cannot fully replace the function of FDX-dependent lipid desaturation.

Highest-during-TAG FADs include SAD1, the ortholog of the soluble plastid desaturase that generates 18:1 fatty acids in plants[61]. In addition, FAD2 is ER-localized and forms the second double bond of C18:2 fatty acids in vitro[58]. Notably, C18:1 and C18:2 are the most highly induced C18 fatty acids during TAG-accumulation in *C. zofingiensis*[58]. In addition, the highest-during-TAG proteins included the ortholog of the *A. thaliana* cytochrome $b_5$ reductase 1 (CBR1, AT5G17770), which contains heme and reduces AtFAD2 (CzFAD2's ortholog) in the ER membrane[57]. Thus, the highest-during-TAG FADs and their electron donors likely generate the major C18 fatty acids in *C. zofingiensis* TAG.

Thylakoid lipids such as MGDG and DGDG contain fatty acids with higher levels of desaturation than storage lipids[56]. Thylakoid loss in heterotrophy occurred concurrently with decreased levels of most FADs (Figs. 2c, 8e). These include FADs implicated biochemically in MGDG and DGDG desaturation (FAD6A, FADA/FAD4, and FAD3A)[58]. These WT−Fe+Glc repressed enzymes were often highest in mixotrophic WT+Fe+Glc cells, which maintain thylakoids, consistent with their participation in thylakoid biogenesis (Figs. 8, 9).

The subset of maintained iron-cofactor proteins during TAG accumulation contrasts with studies that consider a decline in FDX as a universal biomarker of iron deficiency in plants and phytoplankton[25,39]. In *C. reinhardtii*, several FDXs (PETF, FDX3, FDX6) and FADs (CrFAB1, which is CzSAD1's ortholog) are downregulated by iron limitation[32,33,56]. However, while *C. reinhardtii* accumulates TAG during growth on acetate and −Fe, it does so by membrane remobilization rather than *dn*FAS[9,56]. Therefore, FDXs and FADs are unnecessary for *C. reinhardtii* TAG accumulation, but would be required in *C. zofingiensis*, where TAG accumulation depends on *dn*FAS upregulation. Therefore, *dn*FAS is an evolutionary distinct iron priority due to the mechanism of *C. zofingiensis* TAG accumulation (Fig. 9) and provides genetic candidates for synthetic biology and bioengineering in other organisms.

In addition to tracking iron prioritization, our *C. zofingiensis* proteomics allowed us to identify uncharacterized photosynthesis-associated proteins. In plants, systems biology has tracked trophic state transitions including photosynthesis development during plant de-etiolation[30,34,46] and photosynthesis loss during fruit ripening[62]. Unlike these systems, the photosynthetic switch of *C. zofingiensis* is reversible and relies on synergistic nutrient treatments[12,13]. These features enabled us to distinguish candidate photosynthesis-associated proteins (i.e., those that were lowest-in-heterotrophy in −Fe+Glc conditions) from all other proteins that simply respond to a nutrient input per se. Future *C. zofingiensis* mutants that do not repress photosynthesis in +Glc could also be analyzed by proteomics to validate more mechanisms of photosynthetic regulation[63] and functional analysis of these candidate photosynthesis-associated proteins may inspire new strategies to engineer photosynthesis.

Given that photosynthesis switched off in the heterotrophic state, it is notable that the abundance of some annotated photosynthesis proteins barely changed across trophic states. For example, ferredoxin-NADPH reductase (FNR1) remained in the 99th percentile of all protein abundances in both heterotrophic and photosynthetic conditions. By comparison, PsaC (subunit of PSI) decreased from the 98th percentile in WT+Fe−Glc to the 13th percentile in heterotrophy (WT−Fe+Glc). FNR1 likely functions in heterotrophy despite loss of PSI and FDX1, which provide electrons to NADP+ during photosynthesis. In *Z. mays*, a root FNR isoform catalyzes the reverse reduction of FDX by NADPH, allowing for FDX-dependent redox processes in heterotrophic tissue[64]. Like root FNRs, CzFNR1 during heterotrophy could provide electrons from NADPH to CzFDX2, CzFDX5, or CzFLD1 which may participate in FDX-dependent redox reactions, particularly in the upregulated desaturation reactions of *dn*FAS. The highest-in-heterotrophy CzMDH2[47] or the plastid oxidative pentose phosphate pathway, which was maintained in WT−Fe+Glc, potentially provide non-photosynthetic reducing power during heterotrophy. In another example, we found 14/34 detected chlorophyll binding proteins were not in the lowest-in-heterotrophy group (Supplementary Data 5). In fact, proteins like LHC13 and ELIP4 had high expression in −Fe+Glc, potentially showing a photoprotective function of storing residual chlorophyll during non-photosynthetic conditions. This dataset exemplifies how evaluating predicted protein functions against unexpected expression pattern may enhance functional annotation.

Our evolutionary perspective on iron priorities and synergistic iron and Glc responses elucidates conserved genes associated with photosynthesis, lipid accumulation, and iron trafficking across ~1 billion years of evolution between plants and green algae[65]. In fact, the ancestors of vascular plants occupied similar ecological niches as modern unicellular green algae. These organisms could photosynthesize, but likely exploited exogenous organic molecules in diverse nutrient environments. While sugar signaling in plant photosynthesis regulation is often framed as feedback inhibition, sugar signaling in the first photosynthetic eukaryotes may have evolved to respond to exogenous carbohydrates. Therefore, nutrient signaling networks in single-celled green algae likely inform the evolutionary constraints behind vascular plant metabolic regulation. Such evolutionary frameworks can present future strategies to improve agriculture and biofuel production as society increasingly depends on photosynthesis to support a rapidly growing population.

## Methods

### Strains and growth conditions

*C. zofingiensis* SAG 211-14 was the wild-type strain used in this work. Mutants *hxk1-1* and *hxk1-2* were generated previously by UV mutagenesis that produced eight independently derived mutants[13]. Briefly, UV treatment induced hundreds of SNPs in the mutants, but all eight mutants shared consequential point mutations (e.g., frameshift, indels) within the *HXK1* gene (Cz13g07170). *hxk1-1* and *hxk1-2* were predicted to additionally have 42 and 33 high-impact SNP mutations, respectively, including two mutated genes occurring in both strains, Cz10g06190 (PYK4) and Cz17g07220 (putative aspartyl peptidase). These mutations were confirmed by Sanger sequencing, and their proteins had reduced expression in both *hxk1* strains compared to WT in the proteomics datasets (Supplementary Data 3), a protein response also observed for HXK1 (Supplementary Fig. 5).

Growth conditions for wild-type and mutant *C. zofingiensis* cells used a Proteose medium[13] that was recommended by the stock cultures center that we previously found was sufficient to differentiate the phenotypic and transcriptomic effects of Glc addition and removal[12,13]. We noted that while iron was undefined in this original medium, +Fe supplement did not increase growth in −Glc (Fig. 1c, d and Supplementary Fig. 2b). WT, *hxk1-1*, or *hxk1-2* were grown in 1 L of Proteose medium (in a 1.8 L Fernbach flask) with a 16 h:8 h light:dark cycle at 25 °C and 100 μmol photons m$^{-2}$ s$^{-1}$. At mid-log, cultures were split into 110 mL volumes of medium in covered beakers, and treatments began the following day. Treatment replicate cultures were either untreated (control) or received 35 mM of Glc and/or 10 μM of Fe$^{3+}$ from a chelated pre-made stock solution of ferric chloride EDTA[66]. The samples for proteomics were collected at 84 h. For the larger survey of $F_v/F_m$ responses to various nutrients (Supplementary Fig. 1a, b), stock cultures were grown in constant light at 25 °C and 100 μmol photons m$^{-2}$ s$^{-1}$ and split into 45 mL cultures in beakers.

### Growth and biomass measurements

A Multisizer3 (Beckman Coulter) was used to measure both cell density and cell diameter distribution in each culture. Cultures were diluted either 100- or 200-fold in isotonic solution, and then 100 μL of diluted sample was passed through a 50 μm glass aperture for counting. The ".#M3" output files of the Multisizer3 software were processed by a home-made function available in R code (Data and Code Availability) to extract cell density and volumetric biomass per mL. Volumetric biomass was determined as the sum of cellular volumes according to the volume of a sphere formula (1) $(4\pi/3 * (diameter/2)^3) * (cell\ number)$. Biomass specific growth rate per hour was calculated based on the equation (2) $(\log_{10}(biomass\ @\ 72\ h) - \log_{10}(biomass\ @\ 0\ h))/72\ h$.

### Photosynthesis measurements

Photosynthesis was assayed with chlorophyll fluorescence using a Hansatech FMS2 system or oxygen evolution using an Oxygraph Plus System (Hansatech Instruments)[12,13]. For FMS2, samples were dark acclimated for 30 min (shaking ~5 mL in 15 mL tubes) prior to measurements, and an equal volume of culture was filtered into a glass

fiber filter, which was placed on the FMS2 leaf clip. The maximum quantum efficiency of PSII was calculated as $F_v/F_m = (F_m - F_o)/F_m$. $F_m$ was induced by 0.5 saturating pulse >2000 µmol photons m$^{-2}$ s$^{-1}$. We used raw data to quantify the noise of $F_o$ as four standard deviations (-95% of the noise range) of the values attributed to the baseline chlorophyll fluorescence in response to the measuring light. When $F_v$ was below the $F_o$ noise, $F_v$ was considered not detected (n.d.), assigned $F_v/F_m = 0$ for bar charts and statistics and labeled as a red point on graphs (Fig. 1a).

For oxygen evolution, cells were dark acclimated for >30 min (shaking -1–2 mL in 15 mL tubes) and loaded into the Oxygraph electrode chamber (25 °C with 100 rpm stirring) at equal total cell volume per mL with 40 µM sodium bicarbonate. Dark respiration was measured after the rate of oxygen consumption stabilized after transfer into chamber (-3–5 min). Next, the rate of oxygen production was measured at the growth light intensity (100 µmol photons m$^{-2}$ s$^{-1}$) after reaching steady state (-3–5 min). Oxygen evolution is reported as gross oxygen evolution in the light - oxygen consumption in the dark per mm$^3$ of volumetric biomass. All phenotype processing, visualizations, and statistical techniques are in the published R notebooks (Data and Code Availability, "TrophicState_Phenotype.nb.html").

## Transmission electron microscopy

WT *C. zofingiensis* cells were treated with +/−Fe vs. +/−Glc as described above. At 72 h, 25 mL samples in 50 mL tubes were collected and shipped overnight to Pacific Northwest National Laboratory where they were processed upon arrival at -94 h. High-pressure freezing (HPF) and automatic freeze substitution (AFS), followed by plastic embedding, were used to produce thin sections of cell suspension samples. Cells were pelleted by a brief centrifugation, 5 µL was transferred into an HPF flat specimen carrier and frozen with a Leica EM PACT high-pressure freezer (Leica Microsystems, Inc., Bannockburn, IL) at a typical rate of 1700 °C/s. The pods with compacted frozen cells were transferred under liquid nitrogen to the precooled AFS (EM AFS; Leica), and a protocol for cell fixation, water substitution by acetone, and a gradual warm-up to room temperature was followed (see Supplementary Table 1 in Supplementary Information). The samples were then washed three times in acetone and infiltrated with an ascending series of Spurr's low-viscosity embedding resin (Electron Microscopy Sciences, Hatfield, PA) (25, 50, and 75%, followed by three 100% washes for 120 min each), and cured at 60 °C for 48 h. Polymerized blocks were sectioned to 70-nm thin sections with a Leica Ultracut UCT ultramicrotome, mounted on Formvar-coated 100 mesh Cu TEM grids with carbon (Electron Microscopy Sciences, Hatfield, PA), and post-stained with aqueous 2% uranyl acetate (7 min) and Reynolds' lead citrate (3 min)[67]. Samples were imaged with Tecnai T-12 TEM (FEI) with a LaB6 filament, operating at 120 kV. Images were collected digitally with a 2 x 2K Ultrascan1000 CCD (Gatan).

## Starch and glucose quantification

Starch quantification was carried out using a modified method based on the kit from Megazyme (K-TSTA-100A). Frozen sample pellets 84 h after Fe and Glc treatments underwent two washes in 80% (v/v) ethanol at 85 °C. The insoluble fraction was then resuspended in DMSO and heated at 100 °C for 10 min to enhance starch solubilization. Following the kit protocol, starch hydrolysis and quantification were performed. Each experimental condition was conducted with 2–4 biological replicates and the mean of two technical replicates per sample. Starch concentration was calculated as described in the kit documentation and normalized to volumetric biomass.

For glucose estimation, we used the Abcam glucose assay kit (ab272532). Spent media of 84 h samples was used with the protocol in the kit. Briefly, 5 µL of spent media with 500 µL of the reagent were incubated at 100 °C and absorbance was measured at 620 nm. Concentration of glucose was estimated based on a standard curve.

## Proteomics mass spectrometry

For proteomics, -90 mL of culture was centrifuged at 3148 g for 5 min at RT. The cell pellet was resuspended in PBS buffer and washed three times before being flash frozen in liquid nitrogen. Pellets were shipped in dry ice and stored at −80 °C. The protocol for proteomic mass spectrometry was conducted as published in ref. 36. We additionally provide a complete description of off-line fractionation protocols, tandem mass tag labeling, and conditions for mass spectrometry in Supplementary Information. Protein identities and abundances were quantified from all LC-MS/MS spectra using the software MSGF + [68], and reporter ion abundances were extracted from each spectrum using MASIC[69] to measure peptide abundance. Each peptide abundance was normalized to the mean central tendency of all peptide abundances per sample. Full protein abundances were also calculated from the summation of all peptide abundances, with each final protein abundance being normalized to the mean central tendency abundance of each sample (Supplementary Data 3). Downstream statistical analysis of biological replicates (see below) was conducted on log$_2$ abundance of each sample, although samples were de-logged before calculating the mean of biological replicates. Two samples (*hxk1-1 − Fe* +Glc, IDs: 37 and 40) showed strong evidence of incomplete isobaric tag labelling and were removed from downstream statistical analyses.

PC analysis was run on proteins that were detected in every sample and had >2 unique peptides detected using the R package "FactoMineR" (v.2.4). The first two PC coordinates were plotted through ggplot2 (v3.3.5). Autocorrelation was plotted with the package "corrplot" (v.0.92).

## Model of the effects of Fe, Glc, and strain on protein abundance

The complete pipeline and steps for modeling normalized protein abundance as a function of nutrient treatments and genotype, and further process linear models to categorize proteins into unique equation groups is in the published R notebook (Data and Code Availability, TrophicState_Proteomic_Modeling.nb.html) with detailed commentary. To describe the pipeline briefly:

In **STEP1**, proteins that were detected in some but not all 12 conditions were imputed with a value of half the abundance of the least abundant treatment where the protein was detected (Imputed = log$_2$(min(abundance)) -1). Additionally, ≥2 unique peptides per protein had to be detected for statistical analysis.

In **STEP2**, each protein's log$_2$ abundance per condition was fit to the linear model equation below with the "lm" function from the base R "stats" (v.4.1.1) package:

$$\log_2(abundance)_{Fe,Glc,strain} = \alpha + \beta_1\text{Fe} + \beta_2\text{Glc} + \beta_3\text{strain} + \beta_4\text{Fe} : \text{Glc} \\ + \beta_5\text{strain} : \text{Glc} + \beta_6\text{strain} : \text{Fe} + \beta_7\text{strain} : \text{Fe} : \text{Glc}$$

(3)

where $\alpha$ is the abundance intercept per protein and $\beta_x$ is a numerical variable per each effect. Any variable (e.g., *"Fe","Fe:Glc"*) that has a numerical multiplier ($\beta_x$) was called a term in the equation. An equation group refers to the total number of significant terms that describes an equation (e.g., the "Fe, Glc" equation group is dominantly impacted by those two terms). Subgroups indicate the highly correlated sets of proteins where each term describing the groups has a negative or positive effect on the protein. For example, "−Fe+Glc" and "+Fe−Glc" are put in the same equation group, but they are different subgroups with opposite nutrient responses. The "strain" effects were modeled as the difference between WT and *hxk1-2*. The categorical variables +Fe, +Glc, and WT were assigned the values + 1 vs. −Fe, −Glc, and *hxk1* were assigned -1 in the linear model. The intercept ($\alpha$) of each linear model was nearly equivalent to the mean abundance of each protein across all conditions.

In **STEP3**, three numerical criteria were used to establish high confidence in the significance and dominance of a term impacting

protein abundance: (1) any term that had a *p* value greater than Benjamini-Hochberg 5% FDR was removed due to lack of significance. (2) If all remaining terms had $|\beta_x| < 0.25$, no terms were assigned significant impact on abundance. Remaining proteins had their significant terms ordered by the size of each $|\beta_x|$. (3) Per protein, if the $|\beta_x|$ of a given term was ≥4 times larger than any smaller term, all smaller terms were removed due to lack of a dominant effect on protein abundance. For example, the equation "*4Fe + 4Glc + 0.25strain + 0.1strain:Glc*" would be reduced to "*4Fe + 4Glc*". Criterion (3) reduced the overfitting of equation terms that technically had statistical significance but had relatively low impacts on protein abundance. The parameters chosen above were finalized by observing the grouping of well-characterized biomarkers for Fe deficiency[37,39].

**STEP4** iteratively combined subgroups if the mean normalized abundances of two groups had a Pearson correlation >0.9. The combined subgroups were mostly reassigned the name of the larger subgroup (see R notebook for naming exceptions). This final step mostly impacted subgroups with low numbers of proteins that were assigned complex multi-term equations. The scaled abundances of the 12 largest simplified equation groups were plotted as heatmaps with homemade R functions using the "geom_tile" function of ggplot2.

The residuals of the full models showed that ~89% of the residuals are within a 0.5 log$_2$ difference while 99% have a log$_2$ difference <1. An additional assessment of the quality of the modeling fit was conducted through a cross-validation approach. A reduced "*Fe + Glc + Fe:Glc*" model was tested only in WT protein abundance following **STEPS1-3** detailed above, to ensure that these model simplification steps do not lead to overfitting. Then the residuals of the model were determined both against WT (training strain) and *hxk1-2* (test strain) protein data. The distribution of residuals was similar for WT and *hxk1-2* for the set of proteins the original full model simplification labeled as strain-independent (ie. no equation term containing *strain* parameter; Supplementary Fig. 4c). However, a higher frequency of extreme residuals were found in *hxk1-2* compared to WT for proteins the original model labeled as strain-dependent (Supplementary Fig. 4c).

Protein subgroups with >25 proteins were then subjected to GO enrichment analysis to observe specific functions enriched in groups (see below). All protein abundances, sample descriptions, both full linear models and simplified linear effect models, and group-specific GO-term enrichments are available in Supplementary Data 3.

### Enrichment analysis in heterotrophy and TAG accumulation

A set was considered lowest/highest-in-heterotrophy if proteins were significantly lowest or highest in the WT−Fe+Glc condition. Mean lowest/highest-in-heterotrophy were filtered from imputed proteomic abundances that had ≥2 unique peptides per protein with the base R "rank" function. Then, the mean log$_2$ fold difference in abundance was calculated between the lowest or highest abundance in WT−Fe+Glc and the next lowest or highest condition, respectively. Finally, for each protein, a two-tailed *t*-test was run comparing the abundance in WT−Fe+Glc and the abundances in each of the 11 other conditions. The highest *p* value of *t*-tests per protein was derived to ensure that protein abundance of WT−Fe+Glc passed a threshold of significance in each pairwise comparison. The high-stringency significant proteins in heterotrophy had to pass a log$_2$ fold change abundance of 0.5 and have a maximum pairwise *p*-value below the Benjamini-Hochberg 5% FDR. A lower stringency set of proteins included the high-stringency list with the addition of all lowest/highest proteins that passed a 5% FDR. In addition, the lower stringency group added any protein in described by equations where abundance tends to be lowest or highest in −Fe+Glc for both WT and *hxk1-2* (see TrophicState_Proteomic_Modeling.html and Supplemental Data 4 for all considered equations).

The method was nearly identical to find lowest/highest in TAG proteins, with some additional criteria. Both WT−Fe+Glc and WT+Fe+Glc had to be the two least or most abundant conditions, and both log$_2$ fold-difference values and *t*-tests were run with these two TAG-accumulating conditions considered as a single group compared to the other 10 conditions (Supplemental Data 8). Proteins that were lowest or highest in TAG conditions by 5% FDR significance but not by fold change cutoffs were considered less stringent TAG associations. To detect the significance of overlap with the *C. zofingiensis* lipid droplet proteome, Fisher's Exact Test was run for all highest-during-TAG proteins also detected in the published proteomes from ref. 55, using proteins that had ≥2 average spectral counts.

### Gene Ontology (GO) enrichment and predictions of annotations

The designation of GO terms per protein was derived by use of the eggnog mapper function (v2) on the v5.3 proteome of *C. zofingiensis*[36]. GO enrichment was calculated via "topGO" (v1) in R across ontology types using Fisher's Exact Test in R. Exact parameters and homemade functions can be found in TrophicState_Proteomic_Modelling.html (Data and Code Availability).

### Predicting orthology and ortholog enrichment across species

Orthology was determined using Orthofinder2[70] with v5.3 proteome annotations[36] and the following latest versions downloaded from Phytozome: Araport11 (*A. thaliana*), *C. reinhardtii* (v5.5), *Zea mays* (v4), *Oryza sativa* (v.7.0), and *Solanum lycopersicum* (ITAG2.4). The genome annotations of *Nicotiana sylvestris* that were used to align RNA-seq reads by ref. 30 were manually downloaded from an NCBI archival database, and CDS were converted to proteome fasta. Proteins involved in specific pathways were annotated based on the prediction of orthology to the curated proteins in *A. thaliana* and *C. reinhardtii* and occasionally by eggnog emapper results. For respiration, Orthofinder2-produced orthologs were manually assessed for their agreement between *C. zofingiensis* and the *A. thaliana* and *C. reinhardtii* annotated mitochondrial respiratory subunits[71]. Similar ortholog annotations of organellar iron transporters was based on functional predictions of *C. reinhardtii*[49] or *A. thaliana*[22]. Occasionally, the integration of *A. thaliana* and *C. reinhardtii* predictions revealed that *C. zofingiensis* had an *A. thaliana*-typical version of a protein that was absent in *C. reinhardtii* according to gene tree structure. For instance, *C. zofingiensis* has two proteins for PIC1: one ortholog to *A. thaliana* and a separate ortholog to the putative version in *C. reinhardtii*[49]. With *C. zofingiensis* added as an ortholog reference, it appears that *C. reinhardtii* lacks the plant-like PIC1 transporter. The photosynthetic ETC, Cavin-Benson cycle, and chlorophyll biosynthesis genes came from previously annotated homologs[11,12]. Photosynthesis-associated and experimentally validated *A. thaliana* genes were extracted from the "ATH_GO_GOSLIM.txt" file (updated 2022-01-01) from https://arabidopsis.org/download/index-auto.jsp?dir=%2Fdownload_files%2FGO_and_PO_Annotations%2FGene_Ontology_Annotations. All *C. zofingiensis* annotations, with a record of their manual or automated source, used in this publication are available in Supplemental Data 9, which also lists ortholog relationships of proteins with *A. thaliana* and *C. reinhardtii*.

To test whether *C. zofingiensis* lowest-in-heterotrophy proteins were significantly more likely to have an ortholog in a photosynthesis-associated gene list in *C. reinhardtii*, *A. thaliana Z. mays*, or *O. sativa* databases, we developed a Fisher's Exact Test-based method detailed in the R notebook OrthologSignificanceTutorial.nb.html (Data and Code Availability). Briefly, the code takes *C. zofingiensis* lowest-in-heterotrophy proteins that have orthologs in each comparison species (determined by Orthofinder2) and counts the orthologous groups with at least one representative gene/protein in the comparison species' photosynthesis list and tests the significance of overlap compared to a ortholog overlap that would

be expected by chance according to Fisher's Exact Test. The function was run separately on the lower and higher stringency lowest-in-heterotrophy list in *C. zofingiensis* and was compared to mutant libraries in *C. reinhardtii*[5,7], the GreenCut2 database[42], and photosynthesis or tetrapyrrole gene co-expression networks[43]. Conversion of GreenCut2 gene IDs to v5.5 of the *C. reinhardtii* genome from ref. [7] were provided by Setsuko Wakao. Maize genes from the photosynthetic developmental transcriptome[46] were converted from v3 to v4 using the file "B73v3_to_B73v4.tsv" from https://download.maizegdb.org/Pan-genes/B73_gene_xref/. Lists of *A. thaliana* photosynthesis-associated proteins were extracted from the proteomic de-etiolation time course conducted by Ref. [34]. Significance was tested on different fold change cutoffs of $\log_2$ of 0.5, 1, or 2 of de-etiolation at either 24 h (early photosynthetic cotyledon) or 96 h (mature photosynthetic cotyledon) vs. 0 h timepoint (etiolated, heterotrophic cotyledon). Photosynthetic associated maize and rice genes were considered as having $\geq\log_2$ of 1 or 2 mRNA upregulation in three comparable photosynthetic regions per species (M5,M9,M15;R3,R7,R11) compared the base region of the leaf (M1; R1) where cells have undeveloped photosynthetic structures[46].

## Light microscopy

Cell images for each culture condition were taken on using a Zeiss Axios Imager A.2 light microscope attached to a Teledyne Lumenera Infinity5-5 megapixel microscope camera. All photographs were taken from a 63X oil-immersion lens, with the white balance adjusted to the sample background with the Infinity Capture software. Cropping of individual cells was done in ImageJ [v2.0.0-rc-68/1.52 h] where scale bars were added according to a standardized pixel to micron normalization based on a stage micrometer. Relative sizes of .jpegs were maintained for cell size comparison in Adobe Illustrator.

## Thin layer chromatography and high performance liquid chromatography of chlorophyll and carotenoids

For published procedures for visualizing lipids by TLC and measuring carotenoids and chlorophylls by high-performance liquid chromatography[11–13] and the protocol adjustments for this publication, please refer to Supplementary Information.

## Statistics

All statistical analyses were conducted in R. Significance of multiple mean comparisons of phenotypic data (e.g., cell growth, photosynthesis) for time courses and each of the 12 proteomic conditions were performed using the Tukey's Honestly Significant Difference test using the R "stats" TukeyHSD function (0.95 confidence interval). Significantly distinct groups were assigned letter groups using "multcompView" (v.0.1-8) function "multcompLetters" with a $p$ value-adjusted significance threshold of 0.05. Per protein linear modeling was conducted with the following "stats" r lm function: "*lm(abundance ~ strain * Fe *Glc)*" iteratively for each protein. All lowest/highest-in-heterotrophy or in lowest/highest-during-TAG $t$-tests were conducted with the "stats" t.test function, using the default two-tailed calculation. Proteomic heatmaps show each protein in a color scale of $\log_2$ mean abundance of each condition by $\log_2$ mean abundance of that protein across all conditions. After determining the distribution of mean $\log_2(-Fe/+Fe)$ abundances of photosynthetic and respiratory ETCs rejected the null hypothesis of Shapiro-Wilks' test, Wilcoxon Rank Sum Tests was used to test significance of −Fe differences between the two ETC groups. All outputs (e.g., test statistics, $p$ values) of R statistical tests for proteomics and phenotypes can be found in interactive tables in the R notebooks TrophicState_ProteoseTimeCourses.nb.html, TrophicState_Phenotype.nb.html, and TrophicState_ProteomicModeling.nb.html (Data and Code Availability).

## Reporting summary

Further information on research design is available in the Nature Portfolio Reporting Summary linked to this article.

## Data availability

Raw spectra from proteomics has been deposited in the MassIVE database, under subdirectory MSV000092155 [https://massive.ucsd.edu/ProteoSAFe/dataset.jsp?task=46adedf2813f4598baffb6f764c46513]. All source data are provided as a Source Data File and are additionally available in the same repository as the published code (https://osf.io/r8dbe/). Source data are provided with this paper.

## Code availability

The complete pipeline for conducting all analyses and figures, and the input data to generate these, can be found on Open Science Framework (https://osf.io/r8dbe/).

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

## Acknowledgements

We thank Setsuko Wakao for converting the original GreenCut2 gene identifier to *C. reinhardtii* v5 identifiers. This material is based upon work supported by the U.S. Department of Energy, Office of Science, Office of Biological and Environmental Research, under Award Number DE-SC0018301 (M.L, S.S.M, K.K.N, M.S.R). A portion of this research was performed on a project award (DOI: 10.46936/fics.proj.2017.49960/60000021, S.S.M, K.K.N, M.S.R) under the FICUS program and used resources at the DOE Joint Genome Institute and the Environmental Molecular Sciences Laboratory, which are DOE Office of Science User Facilities. Both facilities are sponsored by the Biological and Environmental Research program. K.K.N. is an investigator of the Howard Hughes Medical Institute. This article is subject to HHMI's Open Access to Publications policy. HHMI lab heads have previously granted a non-exclusive CC BY 4.0 license to the public and a sublicensable license to HHMI in their research articles. Pursuant to those licenses, the author-accepted manuscript of this article can be made freely available under a CC BY 4.0 license immediately upon publication.

## Author contributions

T.L.J., K.K.N., and M.S.R. conceptualized and designed experiments for physiology and proteomics. T.L.J., R.M., and M.S.R. conducted algal physiology, analytical chemistry, and culture experiments for proteomics. S.U. and T.J. quantified cellular starch composition and glucose consumption rates. C.D.N and M.L. processed samples for mass-spectrometry. S.O.P. conducted peptide detection and protein quantification from *m/z* spectra with guidance from M.L. A.D. processed and imaged cells by TEM. M.S.R. and S.U. validated *hxk1* strain SNP data and with T.L.J. analyzed strain-specific proteomic response. T.L.J conducted statistical tests of quantified proteins, mined data for evolutionary comparisons, and produced figures with code input from A.S. and manual pathway annotations from K.W. K.K.N. and M.S.R. contributed to data interpretation and editing of figures. S.D.G. and S.S.M. updated *C. zofingiensis* sequence annotations to v5.3. T.L.J. is the primary manuscript author with major contributions from K.K.N. and M.S.R. Relevant methods and supplementary methods sections were written by C.D.N, A.D, S.U., and S.O.P. All authors reviewed and approved the publication.

## Competing interests

The authors declare no competing interests.
