## [Peer Review File · Nature Communications]

REVIEWER COMMENTS

Reviewer #1 (Remarks to the Author):

In the manuscript entitled “Iron rescues glucose-mediated photosynthesis repression during lipid accumulation in the green alga *Chromochloris zofingiensis*” by Jeffers et al., the authors conducted an extensive investigation into the intricate interplay between glucose, HXK1, and Fe in regulating the metabolism of the green alga *C. zofingiensis*. The team untangled the roles of these three factors in shifting the trophic modes of this green alga. Through proteomics analysis, the authors shed new light on the variations in protein abundances across distinct trophic conditions. This study tackles timely research questions and provides compelling evidence to support its conclusions. Certain aspects of the study demand refinement, particularly some methods employed for data analysis. Notably, specific datasets are absent for particular experimental conditions, see details in the following comments:

Figure 1: The plots of normalized chlorophyll fluorescence and delta O₂ nmol are provided for only two conditions, while the remaining panels encompass all four conditions, which raises questions. To enhance transparency, it's recommended that data for -Fe -Glc and +Fe +Glc conditions be added to panels b and d. This data indicates that replete glucose affected (elevated) growth rate more than supplementation of Fe. Were the glucose consumption rates measured for these experiments? Please provide this information.

Throughout the manuscript, volumetric biomass emerges as a primary factor for defining growth and normalizing various characteristics such as O₂ evolution. However, an inconsistency arises in the choice of normalizing factors. For instance, "Total chlorophyll" is normalized by the number of cells in one instance and by volumetric biomass in another. While the explanation provided in lines 212-213 is sound, it would be advantageous to substantiate this approach with either new or previously published data. By looking at the cell images in Fig. 2, it looks like cells in -Fe -Glc may contain a similar amount of starch as cells in +Fe +Glc. This is very critical data for the main outcomes of the manuscript. There is also some inconsistency with other parts that need clarification. For example WT culture: Photosynthetic efficiency under -Fe +Glc is almost negligible compared to +Fe +Glc condition in Fig 1 and Fig 4; the same trend is shown for Total chlorophyll in Fig. 4. However, Total chlorophyll under +Fe +Glc is as low as in -Fe +Glc condition - this is not in agreement with one of the main conclusions of this work.

Proteomics data: the Principal Component Analysis (PCA) reveals well-separated treatment groups; however, a notable lack of distinction between WT and mutant groups raises questions. Providing an explanation for this observation is imperative. On a broader note, the downstream analysis of the proteomics data appears to rely heavily on fitting to linear model equations. It is

difficult or almost impossible without further validation to assess if the data is overfitted or generated unrealistic abundances of proteins. It simply can be seen in differences between model-predicted/normalized protein abundances ranges in each subsets of the data. The authors might want to ensure, using statistical methods, that conclusions are not a result of overfitting.

Figure 4a, please correct the abbreviation from "Wt" to "WT".

Reviewer #2 (Remarks to the Author):

Here the authors have performed a rigorous and comprehensive set of experiments designed to tease out the intricate relationship between glucose, regulator gene HXK1, and iron in governing photosynthesis vs respiration in gene Alga. The work revealed that multiple effectors have a stronger impact on organism response than individual effectors operating alone (an important consideration for things like drug design). The study also demonstrates that these protein response mechanisms are likely conserved across many plants and alga. The heterotrophic conditions studied enabled the authors to tease apart protein membership in various important biological functions like photosynthesis and the accumulation of TAG. The authors fit a linear model to their proteomics data from these heterotrophic condition studies to determine which conditions exercised the greatest control over protein abundance.

Overall, the results are quite compelling. The study and model are able to tease apart different groups of proteins that behave and respond differentially to the tested conditions. The linear model is extremely simple, yet also very effective at binning the proteins based on their distinctive response patterns. Generally the studies are comprehensive, rigorous, well documented, and well described. The writing is excellent. The work is exciting. The authors have established an approach for studying protein function genome-wide using mixotrophic growth that could be replicated in many other systems, and in the process of doing so, they learned a great deal about the roles all the proteins in this particular species play in photosynthesis, energy storage, and respiration. The role iron resource sharing between competing pathways that both have iron requirements plays in dictating how the cell selects its energy strategy is deeply enlightening. This reinforces a growing view that resource allocation is a major driving design consideration in cell regulation. Essentially, this unveils a fundamental design principle for life.

I have only a few suggestions, which I leave to the authors' discretion:

1.) The protein expression heatmaps in figure 3e are tough to interpret. That said, I don't know that there is a better way of doing this. The current representation is very compact, concise, and information rich.

2.) I'd like to have a little more information on how much data agreed with expectations vs being surprising. In the discussion, the authors raise FNR1 as surprising for not changing its expression profile despite being in subsystems that general did change in the tested conditions. How many other proteins were like this? It would be interesting to see a break down of how many genes stayed constant despite being involved in one of the differentially expressed subsystems, how many genes change expression despite being in static subsystems, and so on. This speaks to a potential boon from this work in the form of improved annotations.

3.) Proteomics data not only provides protein abundance information, but also information on many post-translational modifications. Is there evidence of shifts in PTMs across the studied conditions? Particularly for proteins with abundances that stayed constant?

Reviewer #3 (Remarks to the Author):

In their paper titled “Iron rescues glucose-mediated photosynthesis repression during lipid accumulation in the green alga *Chromochloris zofingiensis*”, Jeffers and colleagues explore the response of photosynthesis in the green alga *Chromochloris zofingiensis* to the presence of glucose, and its dependence on iron supplementation. This work included the characterization of the physiological response of these algae to a combination of iron and glucose availability treatments, and was also complemented with studies on mutants of Hexokinase-1 encoding gene. The main dataset generated in this work is the proteome response of WT and HXK1 mutants (one of them might still demonstrate some expression) to these combined treatments, confirming the previously reported physiological strategy of *Chromochloris zofingiensis*, including dnFAS under - Fe+Glc conditions. These proteomes are further studied for novel genes associated with photosynthesis which are downregulated, and for better understanding of iron transport at the sub-cellular level, under these conditions. Finally, Jeffers and colleagues discuss means by which targeting FDX and FAD encoding genes in *C. zofingiensis* might be beneficial for biofuel-directed synthetic approaches.

I found this work interesting, though the role of iron in the carbon response has been characterized in other model green algae. In addition, there are some issues which require clarification or I think must be clarified, some of which are major and may limit the interoperation that can be made based on this work, including the choice of (rich, peptone-based) medium, extremely low levels of iron in the medium, and the dual role of HXK1 which cannot be fully dissected.

Detailed points:

It is unclear to this reviewer why the authors chose a rich medium for an experiment testing effects of glucose addition on PS activity. The rich medium provides a background 'noise' for the heterotrophic conditions elicited by Glc addition. This may be related to the fact that no positive net PS rates are reported in figure 1d (maybe they appear in the -glc controls but they are not presented). Incidentally, in my eyes net rather than gross PS rates should be the values presented for various reasons (the nonlinearity of respiration in dark and light conditions is one) both in general, and especially in the question at hand.

Line 110-112 - mixotrophy is highly unlikely to explain the 4-fold increase in biomass in the +Fe+Glc in view of the negative slopes of net PS as reported in Figure 1d. Moreover, such large increase implies that the cells are Fe-starved, and not just limited, which may limit any interpretation of resource allocation between respiration and photosynthesis, as stress response, protein aggregation and longer reversal time of activity may play a role (see e.g. Devadaus et al 2021).

Lines 119-120 - the second part of the sentence may be completely wrong. I agree that -Fe is the primary nutritional limitation in the experiments performed by the authors, but this on its own doesn't mean that '-Fe is uniquely required for Glc-mediated repression of photosynthesis'. To argue that, the authors must first show that they generate the same limitation for all other nutrients (e.g. as shown in Figure 1e for Fe) and then supplement the cells with each of them separately.

Lines 125-127 - to make such claims, the authors need to provide more than 3 images per treatment in Supp_dataset 2. What more, I am not convinced that 'starch was more prevalent inside +Glc plastids' even in the images provided, but this actually supports the authors' claims regarding the metabolic importance of Fe.

Line 142 - HXK1 mutants cannot fully allow 'distinguish general +Glc response..' as some of the 'general' response is not fully HXK1-independent and is also mediated through its (main) role as a metabolic/glycolytic and not just signaling enzyme. I don't think it's only a terminology issue, but also an important point to qualify the limitations of the analysis (e.g. if a specific signaling associated downstream of HXK1 would have been known and targeted, this could have provided the distinction declared here), which is also supported by the PCA where strain identity was only separated in PC4 and PC8.

Line 215-216 - any suggestion for the smaller investment in antennae? Less cost-effective under mixotrophy? More light penetration per biomass with these cells?

Line 340 - I find the terminology "highest/lowest-in-TAG" potentially inaccurate and at a certain point confusing. Can the authors guarantee that TAG accumulation is the only/main process defining the +Glc treatments?

REVIEWER COMMENTS

Reviewer #1 (Remarks to the Author):

In the manuscript entitled “Iron rescues glucose-mediated photosynthesis repression during lipid accumulation in the green alga *Chromochloris zofingiensis*” by Jeffers et al., the authors conducted an extensive investigation into the intricate interplay between glucose, HXK1, and Fe in regulating the metabolism of the green alga *C. zofingiensis*. The team untangled the roles of these three factors in shifting the trophic modes of this green alga. Through proteomics analysis, the authors shed new light on the variations in protein abundances across distinct trophic conditions. This study tackles timely research questions and provides compelling evidence to support its conclusions. Certain aspects of the study demand refinement, particularly some methods employed for data analysis. Notably, specific datasets are absent for particular experimental conditions, see details in the following comments:

Response: We thank Reviewer #1 for positive comments on our research questions, experimental design and conclusions. We appreciate your suggestions and recommendations, and we have added the mentioned datasets.

Figure 1: The plots of normalized chlorophyll fluorescence and delta O₂ nmol are provided for only two conditions, while the remaining panels encompass all four conditions, which raises questions. To enhance transparency, it's recommended that data for -Fe -Glc and +Fe +Glc conditions be added to panels b and d.

Response: Thank you for your suggestion. We updated Figure 1 with all treatments showing raw data for chlorophyll fluorescence. Because of space, we moved delta O₂ data to the supplement and show data for WT and mutants (Supplementary Fig. 6d).

This data indicates that replete glucose affected (elevated) growth rate more than supplementation of Fe.

Response: Thank you for highlighting this point. We agree and clarified this in the manuscript in lines 112-113.

Were the glucose consumption rates measured for these experiments? Please provide this information.

Response: Thank you for the suggestion. We have added glucose consumption data to the manuscript in lines 110-111 and to the methods in 579-582.

Throughout the manuscript, volumetric biomass emerges as a primary factor for defining growth and normalizing various characteristics such as O₂ evolution. However, an inconsistency arises in the choice of normalizing factors. For instance, "Total chlorophyll" is normalized by the number of cells in one instance and by volumetric biomass in another. While the explanation provided in lines 212-213 is sound, it would be advantageous to substantiate this approach with either new or previously published data.

Response: Thank you for your detailed points on complexities in these data. Because *C. zofingiensis* cell volume increases dramatically both from adding glucose and between -Fe+Glc

and +Fe+Glc (Supplementary Fig. 2a), the normalization is a challenge. We find it logical to look at O₂ evolution normalized by biomass so that one is comparing the same amount of cellular material, and this is what was done in previous publications (Roth et al. 2019 Plant Cell, Roth et al. 2019 Comms Biol). We have updated Fig. 4 to include both total chlorophyll normalizations (Fig. 4e,f) and discussed these data on lines 222-225.

By looking at the cell images in Fig. 2, it looks like cells in -Fe -Glc may contain a similar amount of starch as cells in +e +Glc. This is very critical data for the main outcomes of the manuscript.
Response: Thank you for the suggestion. We added starch concentration measurements to the manuscript in lines 130-132 and Supplementary Fig. 2c as well as the methods in ln 571-578.

Supplementary Figure 2c. Starch at 84 h normalized to volumetric biomass in WT cells.

There is also some inconsistency with other parts that need clarification. For example WT culture: Photosynthetic efficiency under -Fe +Glc is almost negligible compared to +Fe +Glc condition in Fig 1 and Fig 4; the same trend is shown for Total chlorophyll in Fig. 4. However, Total chlorophyll under +Fe +Glc is as low as in -Fe +Glc condition - this is not in agreement with one of the main conclusions of this work.

Response: We appreciate your careful attention and have clarified that total chlorophyll per cell is larger +Fe+Glc than -Fe+Glc (Fig. 4e) while total chlorophyll per biomass (Fig. 4f) is similar due to the large increase in cell biomass in +Fe+Glc, likely due to the buildup of starch and lipids (Fig. 2 and Supplementary Fig. 2c). The differences in photosynthesis between WT+Fe+Glc and WT-Fe+Glc, as evidenced by photosynthetic efficiency and oxygen evolution (Fig. 1 & 4; Supplementary Figure 6), can be explained by the difference in the make-up of chlorophyll. A higher ratio of chl *a* / chl *b* in WT+Fe+Glc than WT-Fe+Glc implicates more core photosystem complexes that evolve oxygen vs. antenna complexes (Supplementary Fig. 6), and the proteomics analysis shows significant upregulation of photosynthetic subunits in WT+Fe+Glc vs. WT-Fe+Glc (Fig. 7a). We also emphasize that oxygen evolution normalized by chlorophyll concentration, a common standardization to compare photosynthetic efficiency (Yang et al. 2020), is ~3x higher in WT+Fe+Glc than in any other photosynthetic condition (Supplementary Fig. 6c) in lines 230-232.

We have modified and added text in lines 223-234 which now references all normalizations in Fig 4 and Supplementary Fig. 6.

Proteomics data: the Principal Component Analysis (PCA) reveals well-separated treatment groups; however, a notable lack of distinction between WT and mutant groups raises questions. Providing an explanation for this observation is imperative.

Response: Thank you for highlighting this point. Two key pieces of evidence underly this result: Evidence 1) *hxc*-independent signaling mechanisms (Aguilera-Alvarado and Sánchez-Nieto, 2017) which include increased cell size in our dataset (Supplementary Fig. 2a)—we have clarified this in lines 202-203. Evidence 2), the finding that these mutants were HXK1 knockdown rather than knockout mutants may imply that residual HXK1 activates some +Glc responses—we have clarified this in the manuscript lines 203-206. Additionally, we have added the strain-dependent PCAs in the manuscript Supplementary Fig. 3b and clarified this in the manuscript in lines 163-164.

Supplementary Fig. 3b. Plot of Principal Components (Dim) 4 and 8 which distinguish strains.

Based on these findings, we focused on the lack of photosynthetic repression or TAG-accumulation in *hxc1* strains as controls for gene discovery rather on elucidating the molecular mechanisms of HXK1 in this paper. We have clarified this in manuscript in lines 206-209.

On a broader note, the downstream analysis of the proteomics data appears to rely heavily on fitting to linear model equations. It is difficult or almost impossible without further validation to assess if the data is overfitted or generated unrealistic abundances of proteins. It simply can be seen in differences between model-predicted/normalized protein abundances ranges in each subsets of the data. The authors might want to ensure, using statistical methods, that conclusions are not a result of overfitting.

Response: Thank you for these comments. We have added the data analyses listed below:

To prevent overfitting, we created additional model simplification steps written in the Methods section (“Model of the effects of Fe, Glc, and strain on protein abundance” starting on line 606). Importantly, we relied on cutoffs based on both relative size of coefficient (i.e. slope of each linear term) and not just solely based on meeting a significant *p*-value. This prevents overfitting

by not allowing small but “technically” significant values to inform the protein response by linear model categorization.

We also looked at the difference between the predicted abundance and the actual abundance (i.e. residuals) and found that for residuals of all models that we derived in Fig 3e and Supplementary Data 3, ~89% of the residuals are within a 0.5 \log_2 difference while 99% have a \log_2 difference < 1. This shows the majority of models accurately predict the abundance. We have incorporated this information in lines 650-660 and lines 178-179.

While the results of the full modeling pipeline show accuracy, we additionally conducted a cross-validation to test that our approach does not overfit. As our full model predicts that most protein abundance effects are strain-independent, we tested if our pipeline models are overfit by running the modelling pipeline on protein abundances from one strain (training strain) and observing how well they predict protein abundance in the same conditions as the other strain (test strain). With WT as the training strain, we reran our pipeline testing for the significance of a “Fe + Glc + Fe:Glc” linear model, removing the strain variables as the model is only run on WT. Model simplification steps then proceeded as mentioned in the Methods section, STEPS1-3. For proteins that were found to be strain-independent in the original full modelling pipeline, the distribution of residuals is similar for the training strain (WT) vs. the test strain (*hxx1-2*, histogram left, Supplementary 4c). This follow-up shows that our pipeline does not overfit under relevant strain-independent datasets, as the test data has similar residuals to the training data. However, proteins found to be strain-dependent tend to have more extreme residuals in *hxx1-2* (histogram, right side) while they predict WT data well. This shows that for these proteins the model suffers from the exclusion of strain-dependent effects. The cross-validation shows the replicability of the model simplification when strain effects have insignificant contributions to protein abundance, confirming the data is not overfit, but also show need to incorporate the full model with strain effects to accurately capture strain-dependent responses.

Supplementary Figure 4c. Histogram of the distribution of the residual (difference in \log_2 abundance) of a reduced cross-validation model (Methods). Models were determined for the equation “Fe + Glc + Fe:Glc” in WT and simplified as in the full-model. Residuals are compared in both WT actual data and *hxx1-2* strain data, which was not used to develop the model. Separate facets of the graphs indicate whether the full modeling pipeline determined a given’s protein abundance was independent (left) or dependent (right) on strain.

Figure 4a, please correct the abbreviation from "Wt" to "WT".

Response: Corrected, thank you for catching this.

Reviewer #2 (Remarks to the Author):

Here the authors have performed a rigorous and comprehensive set of experiments designed to tease out the intricate relationship between glucose, regulator gene HXK1, and iron in governing photosynthesis vs respiration in gene *Alga*. The work revealed that multiple effectors have a stronger impact on organism response than individual effectors operating alone (an important consideration for things like drug design). The study also demonstrates that these protein response mechanisms are likely conserved across many plants and *alga*. The heterotrophic conditions studied enabled the authors to tease apart protein membership in various important biological functions like photosynthesis and the accumulation of TAG. The authors fit a linear model to their proteomics data from these heterotrophic condition studies to determine which conditions exercised the greatest control over protein abundance. Overall, the results are quite compelling. The study and model are able to tease apart different groups of proteins that behave and respond differentially to the tested conditions. The linear model is extremely simple, yet also very effective at binning the proteins based on their distinctive response patterns. Generally the studies are comprehensive, rigorous, well documented, and well described. The writing is excellent. The work is exciting. The authors have established an approach for studying protein function genome-wide using mixotrophic growth that could be replicated in many other systems, and in the process of doing so, they learned a great deal about the roles all the proteins in this particular species play in photosynthesis, energy storage, and respiration. The role iron resource sharing between competing pathways that both have iron requirements plays in dictating how the cell selects its energy strategy is deeply enlightening. This reinforces a growing view that resource allocation is a major driving design consideration in cell regulation. Essentially, this unveils a fundamental design principle for life.

Response: We are incredibly grateful for these enthusiastic comments about our work!

I have only a few suggestions, which I leave to the authors' discretion:

1.) The protein expression heatmaps in figure 3e are tough to interpret. That said, I don't know that there is a better way of doing this. The current representation is very compact, concise, and information rich.

Response: Thank you for acknowledging the complexity of these data and encouraging comments on our representation of these data. Based on previous multifactorial studies (e.g. Swift et al. 2019 Fig. 1d), we decided heatmap plotting was the most concise way to simultaneously capture the number of proteins assigned to each equation-term groups alongside expression differences. While there is a lot of information in one figure, we wanted to have these data available so that a researcher interested in a topic outside the focus of this paper could observe regulatory patterns relevant to their research.

2.) I'd like to have a little more information on how much data agreed with expectations vs

being surprising. In the discussion, the authors raise FNR1 as surprising for not changing its expression profile despite being in subsystems that general did change in the tested conditions. How many other proteins were like this? It would be interesting to see a break down of how many genes stayed constant despite being involved in one of the differentially expressed subsystems, how many genes change expression despite being in static subsystems, and so on. This speaks to a potential boon from this work in the form of improved annotations.

Response: Thank you, this was a very thoughtful comment, and it inspired us to provide a systematic investigation of surprising protein expression deviations from expected function. We used automated bioinformatic software that predicts protein domains (Pfam) and functions based on homology. Then we explored their absence in process related enrichment lists (e.g. lowest-in-heterotrophy) or unexpected linear models to narrow down a list of surprising responses (Supplementary Data 5).

As an example, we investigated all 34 detected proteins with a predicted “Chlorophyll A-B binding” domain. From this Pfam annotation, a researcher may assume these proteins would be linked to photosynthetic light harvesting. 20/34 proteins were indeed lowest-in-heterotrophy, consistent with a role in photosynthesis. However, 14/34 had protein regulatory patterns that did not place them in this list. Interestingly observing the linear models of these 14 that have subterms like -Fe (higher in low Fe) or +Glc (higher when Glc is added) show examples where Chlorophyll A-B binding proteins have higher expression in conditions that normally downregulate chlorophyll abundance. LHC13 tends to be highest in -Fe+Glc conditions and ELIP4 (early light inducible protein) tends to be upregulated in -Fe conditions (Supplementary Data 5). While other LHCs and ELIPs are lowest in heterotrophy (14 LHCs and 4 ELIPS), suggesting their role in light harvesting, the remaining LHC and ELIPS might play more of a role in thylakoid stability, stress tolerance, and storage of non-photosynthetic chlorophyll when photosynthesis is downregulated. We have added this analysis in lines 476-481, made the results available in Supplementary Data 5, and hope this type of research may be used to improve functional annotation in other studies, as the reviewer mentions.

3. Proteomics data not only provides protein abundance information, but also information on many post-translational modifications. Is there evidence of shifts in PTMs across the studied conditions? Particularly for proteins with abundances that stayed constant?

Response: Thank you for bringing up this important aspect. While bioinformatics techniques exist to predict PTM from total protein extracts, we believe it is more appropriate to investigate PTMs from samples that have been enriched for a given PTM because PTMs occur at sparse levels. For this study, we ran mass-spectrometry of whole cell extracts with the specific goal of measuring relative protein abundance. We are currently investigating specific PTMs and hope to include these data in future publications.

Reviewer #3 (Remarks to the Author):

In their paper titled “Iron rescues glucose-mediated photosynthesis repression during lipid accumulation in the green alga *Chromochloris zofingiensis*”, Jeffers and colleagues explore the response of photosynthesis in the green alga *Chromochloris zofingiensis* to the presence of glucose, and its dependence on iron supplementation. This work included the characterization

of the physiological response of these algae to a combination of iron and glucose availability treatments, and was also complemented with studies on mutants of Hexokinase-1 encoding gene. The main dataset generated in this work is the proteome response of WT and HXK1 mutants (one of them might still demonstrate some expression) to these combined treatments, confirming the previously reported physiological strategy of *Chromochloris zofingiensis*, including dnFAS under -Fe+Glc conditions. These proteomes are further studied for novel genes associated with photosynthesis which are downregulated, and for better understanding of iron transport at the sub-cellular level, under these conditions. Finally, Jeffers and colleagues discuss means by which targeting FDX and FAD encoding genes in *C. zofingiensis* might be beneficial for biofuel-directed synthetic approaches.

I found this work interesting, though the role of iron in the carbon response has been characterized in other model green algae. In addition, there are some issues which require clarification or I think must be clarified, some of which are major and may limit the interoperation that can be made based on this work, including the choice of (rich, peptone-based) medium, extremely low levels of iron in the medium, and the dual role of HXK1 which cannot be fully dissected.

Response: We are happy you found our working interesting and thank you for pointing out areas to clarify; our manuscript will be improved by your suggestions.

Detailed points:

It is unclear to this reviewer why the authors chose a rich medium for an experiment testing effects of glucose addition on PS activity.

Response: Thank you for bringing attention to the medium. We chose this medium because it was what this organism was grown in when we ordered this strain from the SAG culture collection and what was recommended for its cultivation. The algae grew well in the medium and in fact increasing the iron levels did not change the growth rate or biomass accumulation in -Glc conditions (Fig. 1c,d, Supplementary Fig. 2b). We used this medium in our original publication on this organism for its genome, transcriptome and carotenoid mutants (Roth et al 2017) and then our former studies of its transcriptomics (Roth et al 2019a) and genetics of *hvk1* strains (Roth et al 2019b). Because of the role of nutrient deficiency in algal metabolism, we started experimenting with iron and others nutrients and discovered the role of iron in rescuing +Glc-mediated repression of photosynthesis (this manuscript). As the physiological, morphological, and molecular responses are easily distinguished between +Glc and -Glc in Proteose medium and iron in Proteose medium is sufficient to rescue photosynthesis, we conducted this studies' multifactorial experiment in a Proteose growth medium. We have clarified this information in the manuscript in lines 508-510. We strongly believe that this is an important study to bring to the scientific community and that these proteomes are a valuable resource.

The rich medium provides a background 'noise' for the heterotrophic conditions elicited by Glc addition. This may be related to the fact that no positive net PS rates are reported in figure 1d (maybe they appear in the -glc controls but they are not presented). Incidentally, in my eyes net rather than gross PS rates should be the values presented for various reasons (the nonlinearity of respiration in dark and light conditions is one) both in general, and especially in the question at hand.

Response: Thank you for these comments. We have added net oxygen evolution in the light for WT and mutants (Fig. 4c) and oxygen consumption data in the dark for WT and mutants (Fig. 4d). We also included a representation of raw oxygen measurements in the light and dark for WT and both mutants (Supplemental Fig. 6d). Fig. 4b-c and Supplementary Fig. 6d show minimal if any background ‘noise’ from the -Glc proteose medium.

Line 110-112 - mixotrophy is highly unlikely to explain the 4-fold increase in biomass in the +Fe+Glc in view of the negative slopes of net PS as reported in Figure 1d. Moreover, such large increase implies that the cells are Fe-starved, and not just limited, which may limit any interpretation of resource allocation between respiration and photosynthesis, as stress response, protein aggregation and longer reversal time of activity may play a role (see e.g. Devadasu et al 2021).

Response: Thank you for this comment. We agree that mixotrophy is highly unlikely to cause the 4-fold increase in biomass, which is stated in Lines 113-115. In extreme iron starvation, as in the study by Devadasu et al. 2021, cells have a dramatic reduction in growth (50% in one generation), and the shape of the cell changes to reflect the stress. In contrast, neither of these changes were observed in *C. zofingiensis*. In -Fe-Glc and -Fe+Glc, the cells grow at a rate similar or more (respectively) than cells in +Fe-Glc (Fig. 1c,d), and the cells do not change shape dramatically (Fig. 2). We have added the growth curves for biomass (in log₁₀ normalized form) and added this to the manuscript with Supplementary Figure 2b and in lines 111-113).

Supplementary Figure 2b. Experimental replicates of WT biomass growth curves after iron and glucose treatment.

Lines 119-120 - the second part of the sentence may be completely wrong. I agree that -Fe is the primary nutritional limitation in the experiments performed by the authors, but this on its own doesn't mean that '-Fe is uniquely required for Glc-mediated repression of photosynthesis'. To argue that, the authors must first show that they generate the same

limitation for all other nutrients (e.g. as shown in Figure 1e for Fe) and then supplement the cells with each of them separately.

Response: Thank you for bringing up this specific point. We have corrected this and made our language more precise in the manuscript in lines 122-123.

Lines 125-127 - to make such claims, the authors need to provide more than 3 images per treatment in Supp_dataset 2. What more, I am not convinced that 'starch was more prevalent inside +Glc plastids' even in the images provided, but this actually support the authors claims regarding the metabolic importance of Fe.

Response: Thank you for this comment. We have added quantified starch data to the manuscript in lines 130-132 and Supplementary Fig. 2c. Additionally, we have made TEM datasets available (link: <https://osf.io/r8dbe/>), and this has been noted in the Figure 2 legend.

Line 142 - HXK1 mutants cannot fully allow 'distinguish general +Glc response..' as some of the 'general' response is not fully HXK1-independent and is also mediated through its (main) role as a metabolic/glycolytic and not just signaling enzyme. I don't think it's only a terminology issue, but also an important point to qualify the limitations of the analysis (e.g. if a specific signaling associated downstream of HXK1 would have been known and targeted, this could have provided the distinction declared here), which is also supported the PCA where strain identity was only separated in PC4 and PC8.

Response: Thank you for this comment. We agree that untangling HXK glycolytic and signaling roles is challenging, and we acknowledge there are limitations in these data. Therefore, we use the *hxx1* mutants as controls to improve our predictions of players of photosynthesis and lipid accumulation rather than to elucidate the mechanism of HXK1. We have revised/clarified this in the manuscript in lines 206-209.

Line 215-216 - any suggestion for the smaller investment in antennae? Less cost-effective under mixotrophy? More light penetration per biomass with these cells?

Response: Thank you for these interesting comments. We interpret the cause as due to regulatory competition: In +Fe+Glc, the reaction centers contain iron cofactors and therefore chlorophyll *a* may be thus induced by +Fe at a stronger level than the periphery complexes. +Glc may have the reverse effect of inhibiting periphery complexes more than reaction center complexes in +Fe conditions. We have revised this in the manuscript in lines 228-230. The reviewer's suggestion on increase light penetration is interesting as +Glc causes cell size increase and astaxanthin accumulation, two features that may reduce light penetration to the plastid.

Line 340 - I find the terminology "highest/lowest-in-TAG" potentially inaccurate and at a certain point confusing. Can the authors guarantee that TAG accumulation is the only/main process defining the +Glc treatments?

Response: Thank you for pointing this out. To make our results clearer, we have changed our terminology in the manuscript to "highest/lowest-during-TAG" to avoid applying lipid accumulation is the only upregulated pathway in these cells.

REVIEWERS' COMMENTS

Reviewer #1 (Remarks to the Author):

The authors have diligently addressed all previous comments. I do not have any major comments.

This manuscript provides an honest representation of data related to nutrient-mediated trophic transitions of a microalgae. This study contributes to our understanding of different nutrient cycles in the environment.

Reviewer #2 (Remarks to the Author):

The authors responded to my limited comments with unexpected vigor and, as a result, added interesting new content to the paper. This was wonderful to see. The response concerning PTMs was very reasonable, and I look forward to seeing future work that digs into this critical aspect of the biology of this system. Overall, I am satisfied this work is ready for publication.

Reviewer #3 (Remarks to the Author):

After going over the revised manuscript, I find that the authors have fully addressed my and other reviewers' comments, and I support its publication.

The only small additional comment I have is that I would briefly mention next to the new text in lines 508-510 that iron did not alter growth /biomass accumulation rates in -Glc conditions, pointing to the relevant data displays.